# Singular Proxies for Adaptive Caching in Diffusion Language Models

**Wenhao Sun** [1]  **Rong-Cheng Tu** [§ 1]  **Yifu Ding** [1]  **Zhao Jin** [1]  **Jingyi Liao** [1]  **Yongcheng Jing** [1]  **Dacheng Tao** [§ 1]

## Abstract

While Diffusion Language Models (DLMs) offer a flexible, arbitrary-order alternative to the autoregressive paradigm, their non-causal nature precludes standard KV caching, forcing costly hidden state recomputation at every decoding step. Existing DLM caching approaches reduce this cost by selective hidden state updates; however, they are still limited by (i) costly token-wise update identification heuristics and (ii) rigid, uniform budget allocation that fails to account for heterogeneous hidden state dynamics. To address these challenges, we present SPA-Cache that jointly optimizes update identification and budget allocation in DLM cache. First, we derive a low-dimensional singular proxy that enables the identification of update-critical tokens in a low-dimensional subspace, substantially reducing the overhead of update identification. Second, we introduce an adaptive strategy that allocates fewer updates to stable layers without degrading generation quality. Together, these contributions significantly improve the efficiency of DLMs, yielding up to an $8\times$ throughput improvement over vanilla decoding and a $2$–$4\times$ speedup over existing caching baselines. The code is available at https://github.com/wenhao728/spa-cache.

## 1 Introduction

Diffusion Language Models (DLMs) (Austin et al., 2021a; Dieleman et al., 2022; Li et al., 2022; Gong et al., 2023) have recently emerged as a versatile alternative to the dominant autoregressive (AR) paradigm (Brown et al., 2020), primarily characterized by their ability to decode in arbitrary orders and facilitate the potential of parallel decoding.

By decoupling generation from a fixed left-to-right order, DLMs have demonstrated remarkable potential in handling multi-modality tasks (You et al., 2025; Yang et al., 2025), overcoming the "reverse curse," (Berglund et al., 2024) enhancing diversity (Gong et al., 2025), and scaling effectively across reasoning (Zhao et al., 2025; Chen et al., 2025; Zhu et al., 2025; Liu et al., 2025a). However, a significant performance gap remains: while proprietary industrial prototype models, such as Gemini Diffusion (DeepMind, 2025) and Mercury (Khanna et al., 2025), have begun to showcase ultra high throughput, current open-source implementations (Nie et al., 2025; Ye et al., 2025) continue to suffer from prohibitive inference latency compared to their AR counterparts. This inefficiency is fundamentally rooted in the unpredictable decoding order and the resulting adaptation of bidirectional attention. Consequently, the standard KV-cache (Pope et al., 2023), a cornerstone of efficiency in AR models, is rendered incompatible, as the hidden states of DLMs cannot be simply reused without recomputation to capture the evolving global dependencies.

Recent efforts to address the DLM inference bottleneck have mainly focused on caching with heuristic-based updates, often assuming a locality bias, where only hidden states in the immediate vicinity of recently decoded tokens require updates (Ma et al., 2025; Jiang et al., 2025). Others rely on restrictive block-wise semi-autoregressive decoding (Wu et al., 2025b;a; Song et al., 2025), which compromises the arbitrary-order flexibility of DLMs and sacrifices global dependency variations. A more principled alternative (Liu et al., 2025b) attempts to identify update-critical tokens by monitoring shifts in the Value states; however, this approach remains bottlenecked. First, the identification process is computationally heavy, requiring high-dimensional projection and similarity computations at every step. This overhead often offsets the gains from sparse updates as the generation lengths scale. Second, these methods typically apply a uniform update budget across all layers. As illustrated in Figure 2, the fraction of drifting tokens, which necessitate updates, shows layer-wise heterogeneity. Consequently, a high fixed budget (*e.g.*, 30%) leads to computational redundancy in stable layers, whereas a lower budget (*e.g.*, 20%) triggers performance degradation due to the insufficient updating of highly unstable layers.

To address these challenges, we present **SPA-Cache**, a

[1]College of Computing and Data Science, Nanyang Technological University, Singapore, Singapore. Correspondence to: Rong-Cheng Tu <rongcheng.tu@ntu.edu.sg>, Dacheng Tao <dacheng.tao@ntu.edu.sg>.

*Proceedings of the $43^{rd}$ International Conference on Machine Learning*, Seoul, South Korea. PMLR 306, 2026. Copyright 2026 by the author(s).

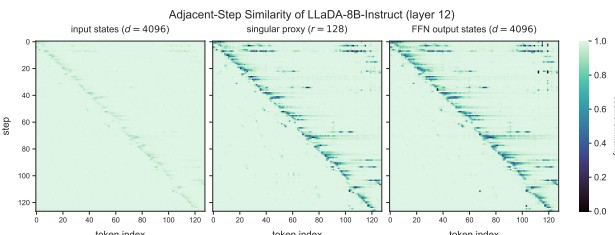

*Figure 1.* **Adjacent-Step Similarities for LLaDA-8B-Instruct** (Nie et al., 2025). While input states exhibit uniformly high similarity, our singular proxy at early stage of the layer can efficiently uncover the drifts in the final FFN output. This provides a clear guidance for selective hidden state updates.

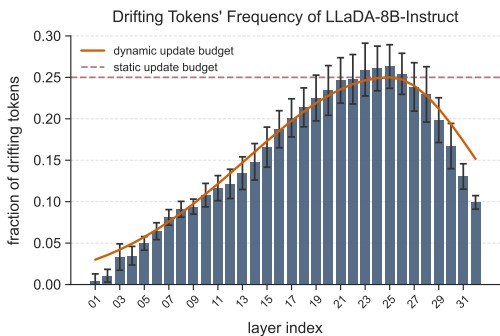

*Figure 2.* **Distribution of Drift Across Layers.** Rather than using a fixed threshold (dashed line) to allocate update budgets uniformly across layers, our SPA-Cache employs a dynamic threshold (solid line) that allocates more updates on the most volatile layers and fewer on the more stable layers.

framework that jointly optimizes update identification and budget allocation. We ground our approach in a formal analysis of the evolutionary dynamics of DLM hidden states, characterizing the statistical properties that dictate when and where updates are most critical. Building on these theoretical insights, we introduce a **singular proxy** that identifies update-critical tokens in the early stages of each layer (Figure 1). Leveraging a singular approximation (Stewart, 1993) of the identification process, our singular proxy retains the dominant directions of state drift in a low-dimensional manifold, thereby avoiding the computational bottleneck associated with high-dimensional identification. Furthermore, observing that hidden state stability fluctuates significantly across model layers, we depart from standard uniform update fractions in favor of an adaptive budget allocation scheme. By dynamically prioritizing computational resources for high-variance layers while aggressively caching stable ones, SPA-Cache maximizes throughput without sacrificing generation quality.

Extensive evaluations on seven benchmarks demonstrate that SPA-Cache significantly optimizes DLM inference, delivering up to a $8\times$ increase in throughput with negligible impact on generation quality. Our SPA-Cache is fundamentally orthogonal to sequence-level optimizations and can be integrated with parallel decoding methods (Wu et al., 2025b) to amplify gains to a total of $28\times$. By drastically reducing inference overhead, this work paves the way and positions DLMs as a computationally viable alternative for high-throughput, real-world applications.

## 2 Related Work

### 2.1 Diffusion Language Models (DLMs)

The transition of diffusion processes to language modeling originated with Gaussian-based continuous latent diffusion in Diffusion-LM (Li et al., 2022) and structured discrete denoising in D3PM (Austin et al., 2021a). Subsequent frameworks such as CDCD (Dieleman et al., 2022) introduced score interpolation to bridge the discrete-continuous gap

between latents and targets. Plaid (Gulrajani & Hashimoto, 2023) demonstrated that diffusion models follow predictable scaling laws, eventually outperforming smaller autoregressive (AR) models in zero-shot likelihood. More recently, MDLM (Sahoo et al., 2024) and LLaDA (Nie et al., 2025) have converged on iterative unmasking as a highly scalable DLM alternative to AR modeling, offering bidirectional context and inherent parallelism. However, the bidirectional nature of these models requires recomputing a forward pass over the entire sequence of length $N$ at each decoding step, resulting in a severe inference bottleneck with $O(T \times N^2)$ complexity for $T$ decoding steps.

### 2.2 Caching Mechanisms for Language Models

KV cache (Pope et al., 2023), a cornerstone of efficiency in AR language models, is fundamentally incompatible with DLMs due to the unpredictable decoding order and the subsequent reliance on full bidirectional attention (Liu et al., 2025b). Recent efforts, like dKV-Cache (Ma et al., 2025), d2Cache (Jiang et al., 2025), and Fast-dLLM (Wu et al., 2025b) propose window-based heuristics that selectively update tokens in the immediate vicinity of recently decoded ones while reusing distant hidden states. However, these methods often rely on a locality bias that lacks theoretical justification and may fail to capture complex global dependency variations. dLLM-Cache (Liu et al., 2025b) monitors feature drift to identify and update critical tokens at arbitrary positions. Despite its flexibility, this approach remains limited by (i) the high overhead of high-dimensional similarity checks and (ii) a uniform update budget across all layers that fails to exploit the layer-wise stability profiles of DLMs.

### 2.3 Other Efficient Generation Techniques

Beyond caching, researchers have explored cache eviction (Song et al., 2025) and specialized attention variants (Wang et al., 2025b; Zhang et al., 2025b) to reduce the cost of

**Algorithm 1** SPA-Cache of a Transformer Block

**input** Input states $\boldsymbol{H}$, cached KV $\boldsymbol{K}^c$, $\boldsymbol{V}^c$, cached FFN outputs $\boldsymbol{H}^c$, layer index $l$, maximum update ratio $\rho_p$
**output** Output states $\boldsymbol{H}$

    **Phase 1: Update Identification & Selection**
    $N \leftarrow \text{len}(\boldsymbol{H})$     ▷ *Sequence length $N$*
    $\rho \leftarrow \text{get\_ada\_rho}(\rho_p, l)$   ▷ *Equation (5) (Section 3.4)*
    $\mathcal{I} \leftarrow \text{get\_update\_indices}(\boldsymbol{H}, k = N \cdot \rho)$  ▷ *Get update indices; Algorithm 2 (Sections 3.2 and 3.3)*
    $\boldsymbol{H}_\mathcal{I} \leftarrow \text{gather}(\boldsymbol{H}, \mathcal{I})$   ▷ $\Pi_\mathcal{I}$*: Select states at indices $\mathcal{I}$;* $\text{len}(\boldsymbol{H}_\mathcal{I}) : k$

    **Phase 2: Attention with Partially Cached KV**
    $\boldsymbol{Q}_\mathcal{I}, \boldsymbol{K}_\mathcal{I}, \boldsymbol{V}_\mathcal{I} \leftarrow \text{to\_qkv}(\boldsymbol{H}_\mathcal{I})$
    $\boldsymbol{K}^c \leftarrow \text{scatter}(\boldsymbol{K}^c, \mathcal{I}, \boldsymbol{K}_\mathcal{I})$
    $\boldsymbol{V}^c \leftarrow \text{scatter}(\boldsymbol{V}^c, \mathcal{I}, \boldsymbol{V}_\mathcal{I})$   ▷ Upd*: Update KV;* $\text{len}(\boldsymbol{K}^c), \text{len}(\boldsymbol{V}^c) : N$
    $\boldsymbol{H}_\mathcal{I} \leftarrow \text{Attn}(\boldsymbol{Q}_\mathcal{I}, \boldsymbol{K}^c, \boldsymbol{V}^c)$

    **Phase 3: FFN and Output States Update**
    $\boldsymbol{H}_\mathcal{I} \leftarrow \text{FFN}(\boldsymbol{H}_\mathcal{I})$
    $\boldsymbol{H}^c \leftarrow \boldsymbol{H} \leftarrow \text{scatter}(\boldsymbol{H}^c, \mathcal{I}, \boldsymbol{H}_\mathcal{I})$   ▷ Upd*: Update output and its cache;* $\text{len}(\boldsymbol{H}) : N$

---

**Algorithm 2** Update Identification

**input** Input states $\boldsymbol{H} = [\boldsymbol{h}_1, \ldots, \boldsymbol{h}_N]$, cached proxy identifiers $\boldsymbol{P}^c = [\boldsymbol{p}_1^c, \ldots, \boldsymbol{p}_N^c]$, update size $k$
**output** Update indices $\mathcal{I}$

    **for** $i = 1$ **to** $N$ **do**
        $\boldsymbol{p}_i \leftarrow f_{\text{proxy}}(\boldsymbol{h}_i)$     ▷ *Compute new identifiers*
        $\boldsymbol{s}_i \leftarrow \mathcal{S}_{\cos}(\boldsymbol{p}_i, \boldsymbol{p}_i^c)$     ▷ *Similarity scores*
    **end for**
    $\boldsymbol{P} \leftarrow [\boldsymbol{p}_1, \ldots, \boldsymbol{p}_N]; \boldsymbol{S} \leftarrow [\boldsymbol{s}_1, \ldots, \boldsymbol{s}_N]$
    $\mathcal{I} \leftarrow \text{topk}(\boldsymbol{S}, k, \text{smallest=True})$   ▷ *Indices to update*
    $\boldsymbol{P}^c \leftarrow \text{scatter}(\boldsymbol{P}^c, \mathcal{I}, \boldsymbol{P})$   ▷ *Update identifiers*

sparse inputs for computation while allowing the remaining tokens to stay dormant and use cache.

**Phase 2: Attention with Partially Cached Key-Value.** The sparse inputs are used to generate a sparse Query $\boldsymbol{Q}_\mathcal{I}$, along with Key $\boldsymbol{K}_\mathcal{I}$ and Value $\boldsymbol{V}_\mathcal{I}$ states exclusively for the tokens indexed in $\mathcal{I}$. The Update (Upd) module then integrates these new KV states into the existing KV cache from previous decoding steps. This process refreshes the representations of the active tokens while reusing cached states for stable tokens. The Attention is computed between the sparse queries and the partially updated KV cache, yielding the sparse attention output.

**Phase 3: FFN and Output States Update.** The final phase handles the feed-forward transitions. The sparse attention output is passed through a FFN component. Similar to Phase 2, an Update (Upd) module uses the indices $\mathcal{I}$ to synchronize the newly computed output states with the cache and produce the final output states for the layer.

This selective recomputation mechanism ensures that the computational complexity of both the Attention and FFN components scales with the number of updated tokens $k$ rather than the total sequence length $N$.

### 3.2 On the Choice of Update Identifier

The core of our framework is the **update identification** mechanism in Phase 1, which selectively determines which tokens require recomputation. As detailed in Algorithm 2, we first project the input hidden states $[\boldsymbol{h}_i]$ into proxy identifier vectors $[\boldsymbol{p}_i]$ via a proxy projection $f_{\text{proxy}}(\cdot)$. To quantify temporal changes, we compute the cosine similarity $\mathcal{S}_{\cos}(\cdot)$ between the current identifier $\boldsymbol{p}_i$ and the cached identifier $\boldsymbol{p}_i^c$ from the previous decoding steps. The update set $\mathcal{I}$ is then constructed by selecting the Top-$k$ tokens with the lowest similarity scores, effectively prioritizing those with the most significant state deviations.

Liu et al. (2025b) has empirically utilized Value states to guide the update of drifting states, *i.e.* $\boldsymbol{p}_i = f_{\text{proxy}}(\boldsymbol{h}_i) = \boldsymbol{W} \boldsymbol{h}_i = \boldsymbol{v}_i$, where $\boldsymbol{W} \in \mathbb{R}^{d \times d}$ is the Value projection

bidirectional modeling. Furthermore, DLMs exhibit unique synergy with parallel decoding methods (Israel et al., 2025; Wang et al., 2025a; Wu et al., 2025b; Lanxiang et al., 2025), where confidence-aware strategies are adopted to simultaneously decode multiple tokens. Caching methods are fully compatible with these parallel frameworks. By integrating our adaptive caching, SPA-Cache, with these parallel frameworks, we can further amplify throughput gains.

## 3 Method

### 3.1 Workflow Overview

As discussed in Section 2.1, the arbitrary-order decoding paradigm and the reliance on bi-directional self-attention in Diffusion Language Models (DLMs) render standard LM caching techniques inapplicable. To address this, we propose **SPA-Cache** for efficient DLM caching. The workflow for a single Transformer layer is partitioned into the following three distinct phases as shown in Algorithm 1 and Figure 3. Some operations (*e.g.*, residual connections, position embeddings) are omitted for clarity.

**Phase 1: Update Identification and Input States Selection.** In this initial phase, the goal is to identify which tokens have drifted significantly to warrant recomputation. Since the drift in subsequent components is unknown beforehand, the challenge lies in determining the update priority efficiently. We employ an update identification Algorithm 2, which will be detailed in Sections 3.2 and 3.3, to select a set of update indices $\mathcal{I}$. These indices guide the selection of

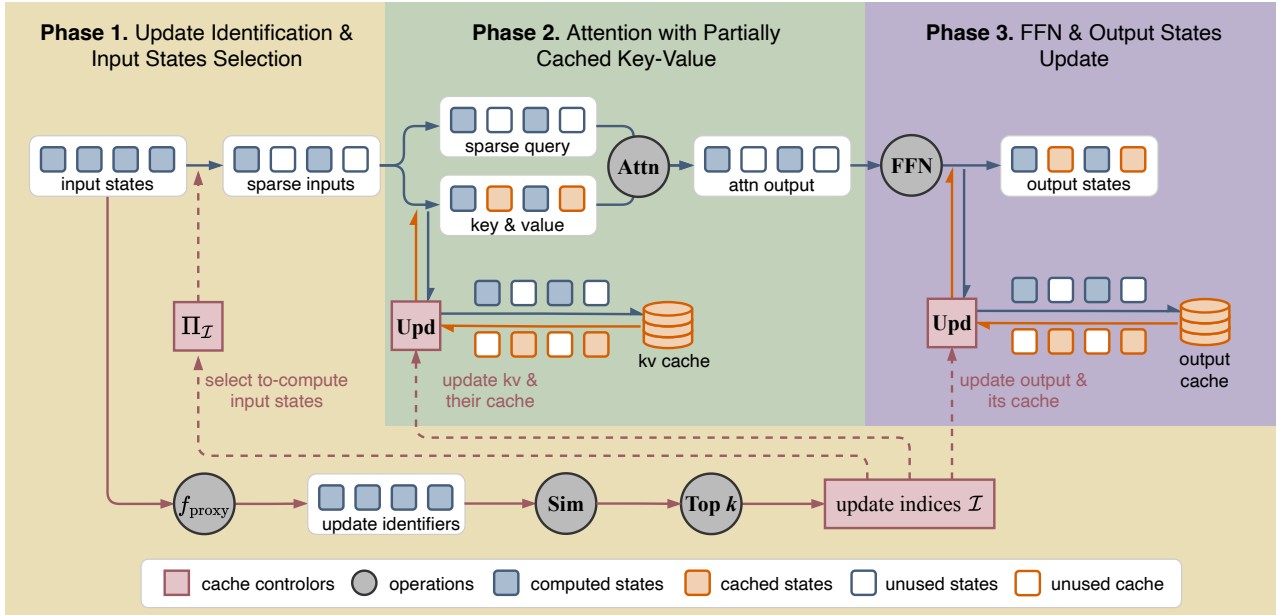

*Figure 3.* **Overview of a Single SPA-Cache Layer. Phase 1**: Input states are projected by $f_{\text{proxy}}$ to produce update scores. A Top-$k$ selection identifies the update index set $\mathcal{I}$, which identifies positions whose cached states from the previous steps should be refreshed. **Phase 2**: Queries are generated only from the selected states and attend to a key-value cache that is selectively updated at indices $\mathcal{I}$. **Phase 3**: The resulting sparse attention outputs are fed into the MLP for the layer output. The output states is updated at $\mathcal{I}$, while non-selected positions reuse cached features.

matrix. However, such design choice was motivated by experimental correlations. In this section, we formalize the theoretical underpinnings of why Value states constitute a principled identifier for state updates. Specifically, we provide analyses of feature drift propagation through linear projections and non-linearities. Full proofs for the following theorems are provided in Appendix A.

We first establish that the stability of the Value states is a sufficient condition for the stability of the attention output.

**Theorem 3.1 (Bound on Attention Output Similarity).** *Consider the Value states $\boldsymbol{v}_i^t \in \mathbb{R}^d$ and attention output $\boldsymbol{h}_i^t \in \mathbb{R}^d$ for the $i$-th token at decoding steps $t$ of a DLM. Under the following assumptions:*

1. *__Bounded Norms:__ The Value norms $\|\boldsymbol{v}_i\|$ and attention output norms $\|\boldsymbol{h}_i\|$ are bounded.*
2. *__Stable Attention:__ The total variation of the attention weights is bounded: $\sum_j |\alpha_{ij}^{t+1} - \alpha_{ij}^t| \leq \delta_A$.*
3. *__Bounded Drift:__ The drift of output states is controlled by its value state: $\sum_j \alpha_{ij}^{t+1}\|\boldsymbol{v}_j^{t+1} - \boldsymbol{v}_j^t\| \leq \lambda\|\boldsymbol{v}_i^{t+1} - \boldsymbol{v}_i^t\|$, where $\lambda \geq 1$ is a relaxation constant.*

*Then, the cosine dissimilarity of the output $\mathbf{h}_i$ is bounded by the dissimilarity of the $i$-th value state $\mathbf{v}_i$:*

$$1 - \mathcal{S}_{\cos}(\boldsymbol{h}_i^t, \boldsymbol{h}_i^{t+1}) \leq C \cdot (1 - \mathcal{S}_{\cos}(\boldsymbol{v}_i^t, \boldsymbol{v}_i^{t+1})) + \epsilon, \quad (1)$$

*where $C > 0$ is a scaling constant determined by the norm bounds, and $\epsilon > 0$ is a residual term capturing attention shifts and norm variance (detailed in Appendix A.1).*

This implies that if the Value states maintain high similarity, the resulting attention output drift is upper bounded. Thus, monitoring Value states provides a conservative guarantee for the stability of the attention output.

Next, we argue that the FFN preserves this stability.

**Theorem 3.2 (Bound on FFN Output Divergence).** *Let $f_{\text{FFN}}(\cdot)$ be a standard FFN component comprising linear projections, RMSNorm (Zhang & Sennrich, 2019), and SiLU (Elfwing et al., 2018) activation functions. Let $\boldsymbol{h}_1, \boldsymbol{h}_2 \in \mathbb{R}^d$ be two FFN input vectors. The divergence of the FFN outputs is bounded by the cosine similarity of their inputs:*

$$\|f_{\text{FFN}}(\boldsymbol{h}_1) - f_{\text{FFN}}(\boldsymbol{h}_2)\|_2 \leq C \cdot \sqrt{1 - \mathcal{S}_{\cos}(\boldsymbol{h}_1, \boldsymbol{h}_2)} + \epsilon, \quad (2)$$

*where $C > 0$ and $\epsilon > 0$ are constants depending on the Lipschitz constant of the network and the feature norms.*

*Remark* 3.3. More generally, this theorem holds for normalizations that project onto a bounded hypersphere (*e.g.*, LayerNorm (Ba et al., 2016)) and Lipschitz-continuous activations (*e.g.*, GELU (Hendrycks & Gimpel, 2016)). The bounding property can also be extended to the more recent MoE (Zhang et al., 2025a) as they are intrinsically composed of standard FFNs.

Theorem 3.1 and Theorem 3.2 together provide the theoretical guarantee for the Value identification mechanism: if the Value states maintain high cosine similarity, the final layer output drifts are bounded. Consequently, it is unnecessary

*Table 1.* Comparison of different identifier types on **LLaDA-8B-Instruct** (Nie et al., 2025) and GSM8K (Cobbe et al., 2021). TPS: Tokens Per Second, TTFT: Time To First Token (in milliseconds).

| IDENTIFIER TYPE | TPS ↑ | TTFT ↓ | ACCURACY ↑ |
|---|---|---|---|
| **BASELINE (NONE)** | 29.01 | 28.6 | 78.62 $_{(\pm1.13)}$ |
| QUERY | 164.36 | 28.4 | 77.21 $_{(\pm1.15)}$ |
| KEY | 164.11 | 28.1 | 76.83 $_{(\pm1.15)}$ |
| VALUE | 164.88 | 28.1 | **78.59** $_{(\pm\mathbf{1.13})}$ |
| ATTN. INPUT | **172.47** | 28.6 | 77.29 $_{(\pm1.14)}$ |
| ATTN. OUTPUT | 139.18 | 28.0 | 73.92 $_{(\pm1.21)}$ |

(and impractical) to speculatively evaluate the whole layer to detect divergence; the similarity of Value states act as a sufficient proxy for determining which tokens can safely reuse cached states.

**Emperical Evaluation of the Identifiers.** To validate our theoretical findings, we perform an empirical comparison of various hidden state components as update identifiers in Table 1. Specifically, we evaluate the efficacy when using Query, Key, Value, attention input, and attention output states as update identifiers. As demonstrated in Table 1, the Value state outperforms all other candidates, providing the most reliable signal for hidden state drift.

Query and Key serve as routing features; while shifts in their values may alter the attention score distribution, the resulting output remains stable if the attended Value vectors are semantically equivalent. The attention input state is an entangled representation of Query, Key, and Value states, making it less effective for isolating drift. Notably, using attention output states as identifiers yields the most significant accuracy degradation. This limitation stems from the Anisotropy Masking Effect (Ethayarajh, 2019; Gao et al., 2019; Timkey & van Schijndel, 2021), in which the representation space of deeper features collapses into a narrow cone. This dominance of a few directions masks the unique features necessary for robust identification. We provide an in-depth analysis of this failure mode in Appendix B.

### 3.3 Singular Proxy Identifier

Although the Value states are justified as a reliable identifier for update selection, projecting the whole input sequence to Values and computing similarity in the $d$-dimensional Value space are computationally expensive. In moderate-parameterized DLMs such as LLaDA-8B ($d = 4096$), the identification process in this $d$-dimensional space incurs an overhead that can offset the gains from sparse computation.

To address this challenge, we introduce the **singular proxy**, a method that designed to identify update-critical tokens within a low-dimensional subspace $r \ll d$. We perform the Singular Value Decomposition (SVD) (Stewart, 1993) on the Value projection matrix $W \in \mathbb{R}^{d \times d}$ such that

$W \approx U\Lambda V^\top$, where $U, V \in \mathbb{R}^{d \times d}$ denote the left and right singular vectors, and $\Lambda = \mathrm{diag}(\lambda_1, \ldots, \lambda_d) \in \mathbb{R}^{d \times d}$ represents the singular values in descending order. We define a truncated projection $W_r \in \mathbb{R}^{r \times d}$ by retaining the top $r$ right singular vectors and singular values. This yields the dimension-reduction projection, $f_{\mathrm{proxy}} : \mathbb{R}^d \to \mathbb{R}^r$:

$$f_{\mathrm{proxy}}(h_i) = W_r h_i = (\Lambda_r V_r^\top)h_i, \tag{3}$$

where $\mathbf{h}_i \in \mathbb{R}^d$ is the $i$-th input state. By projecting the input into the principal subspace, we reduce the computational complexity of the identifier mapping $f_{\mathrm{proxy}}(\cdot)$ from $O(d^3)$ to $O(rd^2)$ and the similarity computation $\mathcal{S}_{\cos}(\cdot)$ from $O(d)$ to $O(r)$. The remaining pipeline still follows the procedure outlined in Algorithm 2.

While truncated SVD preserves the majority of variance, it is critical to ensure that the topological structure remains intact within the reduced subspace. To formalize this, we establish the following bound on similarity divergence:

**Theorem 3.4** (**Similarity Preservation in the Truncated Projection**)**.** *Let* $\boldsymbol{h}_1, \boldsymbol{h}_2 \in \mathrm{span}(V_r)$ *be the input states. Denote* $\boldsymbol{v} = \boldsymbol{W}\boldsymbol{h}$ *and* $\hat{\boldsymbol{v}} = \boldsymbol{W}_r\boldsymbol{h}$ *as the Value states and our singular proxy, respectively. Their divergence of the cosine similarity is bounded by a factor independent of the input states* $\boldsymbol{h}_1, \boldsymbol{h}_2$:

$$|\mathcal{S}_{\cos}(\boldsymbol{v}_1, \boldsymbol{v}_2) - \mathcal{S}_{\cos}(\hat{\boldsymbol{v}}_1, \hat{\boldsymbol{v}}_2)| \leq 2\left(\frac{\lambda_{r+1}}{\lambda_r}\right)^2. \tag{4}$$

*Remark* 3.5. The $\boldsymbol{h}_1, \boldsymbol{h}_2 \in \mathrm{span}(V_r)$ assumption typically holds as the cumulative update to each neuron can be viewed as a weighted summation of the training input vectors (Cheng et al., 2025). The rigorous proof is provided in Appendix A.3.

By leveraging Theorem 3.4, we can analytically ensure the singular proxy is an effective alternative for the Value proxy. This allows for computational reductions during the update identification process.

### 3.4 Adaptive Budget Allocation

While the singular proxy identifier in Section 3.3 reduces per-token identification overhead, the layer's computation is also governed by the update ratio $\rho$ (*i.e.*, the fraction of tokens updated). Prior method (Liu et al., 2025b) typically employs a uniform budget allocation, maintaining a static update ratio (*e.g.*, $\rho = 0.25$) across all layers. However, our analysis of hidden state evolution reveals significant layer-wise heterogeneity that renders uniform allocation less efficient.

As shown in Figure 2, we evaluate the average ratio of highly drifting tokens per layer (defined as tokens whose cosine similarity with last decoding step falls below a threshold $\tau =$

*Table 2.* Comparison of LLaDA-8B-Instruct (Nie et al., 2025) and Dream-v0-Instruct-7B (Ye et al., 2025) across seven benchmarks. Compared to prior dLLM-Cache (Liu et al., 2025b) and Fast-dLLM (Wu et al., 2025b), our SPA-Cache achieves better speedups while maintaining comparable accuarcy.

| TASK | METHOD | LLaDA-8B-Instruct | | | Dream-v0-Instruct-7B | | |
|---|---|---|---|---|---|---|---|
| | | TPS ↑ | TTFT (MS) ↓ | ACCURACY ↑ | TPS ↑ | TTFT (MS) ↓ | ACCURACY ↑ |
| | | MATHEMATICS & SCIENCE | | | | | |
| GSM8K | BASELINE | 29.67 $_{(1.0\times)}$ | 28.0 | 78.62 $_{(\pm1.13)}$ | 17.86 $_{(1.0\times)}$ | 23.6 | 75.21 $_{(\pm1.19)}$ |
| | + dLLM-CACHE | 68.62 $_{(2.3\times)}$ | 28.7 | 78.24 $_{(\pm1.14)}$ | 23.39 $_{(1.3\times)}$ | 25.3 | 74.45 $_{(\pm1.20)}$ |
| | + FAST-dLLM | 93.86 $_{(3.2\times)}$ | 28.1 | 76.80 $_{(\pm1.16)}$ | 25.53 $_{(1.4\times)}$ | 24.6 | 79.23 $_{(\pm1.12)}$ |
| | + OURS | **190.73** $_{(6.4\times)}$ | 28.7 | 78.24 $_{(\pm1.14)}$ | **45.97** $_{(2.6\times)}$ | 24.5 | 77.56 $_{(\pm1.15)}$ |
| GPQA | BASELINE | 20.55 $_{(1.0\times)}$ | 44.8 | 29.91 $_{(\pm2.17)}$ | 12.89 $_{(1.0\times)}$ | 61.2 | 34.15 $_{(\pm2.24)}$ |
| | + dLLM-CACHE | 27.29 $_{(1.3\times)}$ | 44.9 | 29.01 $_{(\pm2.15)}$ | 25.40 $_{(2.0\times)}$ | 61.4 | 33.04 $_{(\pm2.22)}$ |
| | + FAST-dLLM | 25.91 $_{(1.3\times)}$ | 44.4 | 27.23 $_{(\pm2.11)}$ | 30.63 $_{(2.4\times)}$ | 61.3 | 33.93 $_{(\pm2.24)}$ |
| | + OURS | **39.19** $_{(1.9\times)}$ | 44.6 | 30.58 $_{(\pm2.18)}$ | **45.97** $_{(3.6\times)}$ | 61.8 | 34.60 $_{(\pm2.25)}$ |
| MATH500 | BASELINE | 33.35 $_{(1.0\times)}$ | 23.3 | 33.18 $_{(\pm0.62)}$ | 23.08 $_{(1.0\times)}$ | 20.2 | 36.16 $_{(\pm0.63)}$ |
| | + dLLM-CACHE | 74.26 $_{(2.2\times)}$ | 23.2 | 33.14 $_{(\pm0.62)}$ | 42.08 $_{(1.8\times)}$ | 21.9 | 32.86 $_{(\pm0.63)}$ |
| | + FAST-dLLM | 85.94 $_{(2.6\times)}$ | 23.2 | 32.02 $_{(\pm0.62)}$ | 89.05 $_{(3.9\times)}$ | 20.9 | 35.04 $_{(\pm0.63)}$ |
| | + OURS | **172.19** $_{(5.2\times)}$ | 24.8 | 33.44 $_{(\pm0.62)}$ | **104.23** $_{(4.5\times)}$ | 20.8 | 34.84 $_{(\pm0.63)}$ |
| | | GENERAL QA | | | | | |
| BBH | BASELINE | 24.85 $_{(1.0\times)}$ | 16.2 | 53.49 $_{(\pm0.54)}$ | 49.30 $_{(1.0\times)}$ | 14.0 | 58.42 $_{(\pm0.52)}$ |
| | + dLLM-CACHE | 40.48 $_{(1.6\times)}$ | 16.5 | 52.74 $_{(\pm0.55)}$ | 54.65 $_{(1.1\times)}$ | 14.0 | 58.22 $_{(\pm0.52)}$ |
| | + FAST-dLLM | 105.82 $_{(4.3\times)}$ | 17.0 | 54.66 $_{(\pm0.56)}$ | 123.05 $_{(2.5\times)}$ | 13.2 | 57.79 $_{(\pm0.52)}$ |
| | + OURS | **108.31** $_{(4.4\times)}$ | 16.1 | 53.72 $_{(\pm0.55)}$ | **153.61** $_{(3.1\times)}$ | 13.2 | 58.52 $_{(\pm0.52)}$ |
| MMLU-PRO | BASELINE | 20.68 $_{(1.0\times)}$ | 43.8 | 37.08 $_{(\pm0.43)}$ | 10.06 $_{(1.0\times)}$ | 33.9 | 45.01 $_{(\pm0.44)}$ |
| | + dLLM-CACHE | 52.71 $_{(2.5\times)}$ | 43.5 | 36.50 $_{(\pm0.43)}$ | 15.53 $_{(1.5\times)}$ | 34.3 | 44.16 $_{(\pm0.44)}$ |
| | + FAST-dLLM | 81.25 $_{(3.9\times)}$ | 44.1 | 37.28 $_{(\pm0.43)}$ | 50.58 $_{(5.0\times)}$ | 34.2 | 46.62 $_{(\pm0.44)}$ |
| | + OURS | **124.06** $_{(6.0\times)}$ | 44.2 | 36.30 $_{(\pm0.43)}$ | **62.13** $_{(6.2\times)}$ | 35.0 | 45.16 $_{(\pm0.44)}$ |
| | | CODE | | | | | |
| MBPP | BASELINE | 5.75 $_{(1.0\times)}$ | 28.7 | 39.20 | 36.51 $_{(1.0\times)}$ | 22.2 | 57.40 |
| | + dLLM-CACHE | 8.38 $_{(1.5\times)}$ | 28.8 | 39.00 | 50.31 $_{(1.4\times)}$ | 21.9 | 56.20 |
| | + FAST-dLLM | 12.49 $_{(2.2\times)}$ | 28.6 | 37.80 | 58.42 $_{(1.6\times)}$ | 21.7 | 51.00 |
| | + OURS | **46.12** $_{(8.0\times)}$ | 28.7 | 39.00 | **114.85** $_{(3.1\times)}$ | 22.6 | 57.60 |
| HUMANEVAL | BASELINE | 37.48 $_{(1.0\times)}$ | 14.4 | 42.07 | 28.00 $_{(1.0\times)}$ | 14.0 | 58.54 |
| | + dLLM-CACHE | 40.29 $_{(1.1\times)}$ | 14.3 | 39.63 | 21.15 $_{(0.8\times)}$ | 14.1 | 45.12 |
| | + FAST-dLLM | 81.90 $_{(2.2\times)}$ | 14.3 | 41.71 | 40.34 $_{(1.4\times)}$ | 13.8 | 53.05 |
| | + OURS | **132.91** $_{(3.5\times)}$ | 14.3 | 42.07 | **42.49** $_{(1.5\times)}$ | 13.4 | 58.54 |

0.95). This analysis, conducted over 100 random samples from GSM8K (Cobbe et al., 2021), MMLU-pro (Wang et al., 2024), and MBPP (Austin et al., 2021b), demonstrates a consistent distribution across diverse tasks. Specifically, we observe that state drift varies drastically across the model depth: (1) Hidden states remain relatively stable as they primarily perform initial embedding transformations in the early layers; (2) Drift peaks as the model performs complex semantic transformations in the middle layers, requiring more frequent updates; (3) Dynamics stabilize again as approaching the final layer of the model. A fixed update threshold (*e.g.*, $\rho = 0.25$, represented by the dashed line in Figure 2) essentially over-allocates budget to the stable early layers and late layers.

To exploit this heterogeneity, SPA-Cache replaces rigid allocation with an adaptive budget allocation scheme. Noticing that the drift curve roughly follows an asymmetric bell shape, we parameterize the layer-wise dynamic update ratio

$\rho(l)$ using a piecewise Gaussian function:

$$\rho(l) = \begin{cases} \rho_p \cdot \exp\left(\ln\left(\frac{\rho_1}{\rho_p}\right) \cdot \left(\frac{l-l_p}{l_p-1}\right)^2\right) & \text{if } l \leq l_p \\ \rho_p \cdot \exp\left(\ln\left(\frac{\rho_L}{\rho_p}\right) \cdot \left(\frac{l-l_p}{L-l_p}\right)^2\right) & \text{if } l > l_p \end{cases}, \quad (5)$$

where, $l$ denotes the current layer index. $L$ is the total layer nubmer of the model. $\rho_p$ is the peak update ratio at layer $l_p$. And $\rho_1, \rho_L$ represent the boundary update ratios for the first and last layers, respectively. This parameterization provides the flexibility to adapt to various DLMs with different skewness in their profiles. More analysis and configurations are detailed in Appendix C.

The dynamic strategy ensures that computational resources are concentrated on high-variance layers where updates are most critical for preserving accuracy, while aggressively caching stable layers to maximize throughput.

# 4 Experiments

## 4.1 Experimental Setup

**Models and Benchmarks.** We evaluate SPA-Cache across two representative Diffusion Language Models (DLMs): LLaDA-8B (Nie et al., 2025) and Dream-7B (Ye et al., 2025). To assess generalizability, we conduct evaluations on seven diverse benchmarks covering mathematical reasoning (GSM8K (Cobbe et al., 2021), MATH500 (Hendrycks et al., 2021)), scientific knowledge (GPQA (Rein et al., 2023)), general-purpose QA (BBH (Suzgun et al., 2023), MMLU-pro (Wang et al., 2024)), and code generation (MBPP (Austin et al., 2021b), HumanEval (Chen et al., 2021)).

**Baselines.** We compare SPA-Cache against vanilla DLM decoding (no cache) and two state-of-the-art caching frameworks: dLLM-Cache (Liu et al., 2025b) and Fast-dLLM (Wu et al., 2025b).

**Implementation Details.** All experiments are executed on a single NVIDIA B200 GPU. For all experiments, we adopt the configuration from and set the allocation hyperparameter to $\rho_p = 0.25$. We report accuracy (for QA/Reasoning) and pass@1 (for coding tasks). To quantify efficiency gains, we measure average Tokens Per Second (TPS) and Time-To-First-Token (TTFT). Further implementation details and additional results on the post-trained LLaDA-1.5 (Zhu et al., 2025) are provided in Appendix D and E, respectively.

## 4.2 Main Results

Table 2 shows that SPA-Cache consistently outperforms existing caching methods across all benchmarks. By leveraging both the singular proxy identifier and adaptive budget allocation SPA-Cache achieves substantial throughput improvements and maintains comparable accuracy to vanilla decoding. Specifically, on the MMLU-pro benchmark, SPA-Cache yields a $6.0\times$ and $6.2\times$ speedup on LLaDA-8B and Dream-7B, respectively. The most significant gain is observed on MBPP with LLaDA-8B, where SPA-Cache attains an $8.0\times$ throughput acceleration.

## 4.3 Integration with Parallel Decoding.

An advantage of DLMs is their inherent potentiality for parallel decoding (Nie et al., 2025). SPA-Cache is orthogonal to these techniques. To demonstrate this synergy, we integrate SPA-Cache with the parallel decoding framework from Wu et al. (2025b).

As shown in Table 3, combining SPA-Cache with parallelization yields further efficiency gains. While parallel decoding introduces a marginal distribution approximation error (resulting in a slight accuracy trade-off), the throughput improvements are remarkable. On the BBH benchmark, the combined approach achieves an unprecedented $28\times$

*Table 3.* **Integration of SPA-Cache with parallel strategy** (Wu et al., 2025b). Our approach outperforms the Fast-dLLM dual cache in throughput while preserving comparable accuracy.

| TASK | METHOD | TPS ↑ | ACCURACY ↑ |
|---|---|---|---|
| MATHEMATICS & SCIENCE | | | |
| GSM8K | BASELINE | 29.67 (1.0×) | 78.62 (±1.13) |
| | + FAST-DLLM | 176.45 (5.9×) | 77.02 (±1.16) |
| | + OURS | 276.39 (9.3×) | 76.81 (±1.18) |
| GPQA | BASELINE | 20.55 (1.0×) | 29.91 (±2.17) |
| | + FAST-DLLM | 44.40 (2.2×) | 27.68 (±2.12) |
| | + OURS | 47.82 (2.3×) | 27.01 (±2.10) |
| MATH500 | BASELINE | 33.35 (1.0×) | 33.18 (±0.62) |
| | + FAST-DLLM | 86.69 (2.6×) | 31.76 (±0.62) |
| | + OURS | 289.71 (8.7×) | 32.66 (±0.63) |
| GENERAL QA | | | |
| BBH | BASELINE | 24.85 (1.0×) | 51.49 (±0.54) |
| | + FAST-DLLM | 301.33 (12.1×) | 53.20 (±0.56) |
| | + OURS | 693.96 (27.9×) | 53.13 (±0.56) |
| MMLU-PRO | BASELINE | 20.68 (1.0×) | 37.08 (±0.43) |
| | + FAST-DLLM | 86.40 (4.2×) | 37.34 (±0.43) |
| | + OURS | 224.97 (10.9×) | 36.85 (±0.42) |
| CODE | | | |
| MBPP | BASELINE | 5.75 (1.0×) | 39.20 |
| | + FAST-DLLM | 50.11 (8.7×) | 37.80 |
| | + OURS | 143.25 (24.9×) | 38.20 |
| HUMANEVAL | BASELINE | 37.48 (1.0×) | 42.07 |
| | + FAST-DLLM | 117.67 (3.1×) | 39.81 |
| | + OURS | 215.83 (5.8×) | 40.24 |

*Table 4.* Ablation on identifier and adaptive budget allocation.

| IDENTIFIER | PEAK $\rho_p$ | AVG $\bar{\rho}$ | TPS ↑ | ACCURACY ↑ |
|---|---|---|---|---|
| NONE | 100% | 100% | 29.01 | 78.62 (±1.13) |
| VALUE | 25% | 25% | 164.88 | **78.59** (±**1.13**) |
| SINGULAR$_{128}$ | 25% | 25% | 179.43 | 78.23 (±1.12) |
| SINGULAR$_{128}$ | 25% | 16% | **189.13** | 78.24 (±1.14) |
| SINGULAR$_{128}$ | 16% | 16% | **190.06** | 75.65 (±1.17) |

speedup over vanilla decoding. Crucially, SPA-Cache continues to outperform Fast-dLLM's dual cache within the parallelized setting, showcasing its robustness.

## 4.4 Ablations

In this section, we conduct extensive ablation studies to systematically evaluate the individual contributions of our proposed components to the overall performance of SPA-Cache in practaice. Unless otherwise specified, all experiments in this section are performed using the LLaDA-8B model (Nie et al., 2025) on the GSM8K (Cobbe et al., 2021) benchmark.

**Efficacy of Proposed Components.** We isolate the contributions of the singular proxy and adaptive allocation in Table 4. Replacing the Value proxy identifier with our singular proxy identifier (Section 3.3) increases TPS from 165 to 179 without degrading accuracy. Furthermore, introducing the adaptive budget allocation (Section 3.4) allows the

*Table 5.* **Impact of Proxy Rank $r$ on LLaDA-8B-Instruct (Zhu et al., 2025), GSM8K (Cobbe et al., 2021).** While the full-dimensional Value identifier uses $d = 4096$, our singular value proxy achieves comparable performance at $r = 512$. Notably, $r = 128$ strike a good balance between throughput and accuracy.

| IDENTIFIER | TPS ↑ | ACCURACY ↑ |
|---|---|---|
| **NONE** | 29.01 | 78.62 $_{(\pm1.13)}$ |
| VALUE | 164.88 | 78.59 $_{(\pm1.13)}$ |
| SINGULAR$_{512}$ | 172.56 | 78.57 $_{(\pm1.13)}$ |
| SINGULAR$_{256}$ | 176.38 | 78.47 $_{(\pm1.12)}$ |
| SINGULAR$_{128}$ | 179.43 | 78.23 $_{(\pm1.12)}$ |
| SINGULAR$_{64}$ | 181.83 | 77.79 $_{(\pm1.14)}$ |
| SINGULAR$_{32}$ | 182.66 | 76.71 $_{(\pm1.15)}$ |

model to reduce the average update ratio from $\bar{\rho} = 25\%$ to $\bar{\rho} = 16\%$, pushing the TPS to 189. Notably, when we ignore the layer-wise heterogeneity and force a uniform 16% update ratio across all layers, accuracy drops from 78.23% to 75.56%. This confirms that our adaptive strategy successfully identifies and exploits stable layers to save computation without sacrificing model performance.

**Impact of Singular Proxy Rank.** The dimension of the singular proxy ($r$) controls the trade-off between identification overhead and approximation quality. We quantitatively analyze this in Table 5. We observe that while lower dimensions further improve TPS, accuracy begins to decline when $r < 128$. We therefore select $r = 128$ as the sweet spot and use it as the default value across all experiments.

**Analysis of Computational Overhead.** To investigate the source of our efficiency gains, we profile the per-layer breakdown for LLaDA-8B in Figure 4 (setting prompt length 1024, decoding length 256, and $\rho = 5\%$). In the vanilla baseline, execution time is dominated by the linear projections and non-linearities in the Attention and FFN components. While the Value proxy caching methods successfully bypass these primary costs, they introduce a non-trivial identification overhead that emerges as a new computational bottleneck. In contrast, SPA-Cache's singular proxy enables token selection in a highly compressed subspace, drastically reducing identification latency. This optimization allows the total execution time to approach the theoretical floor, maxi-

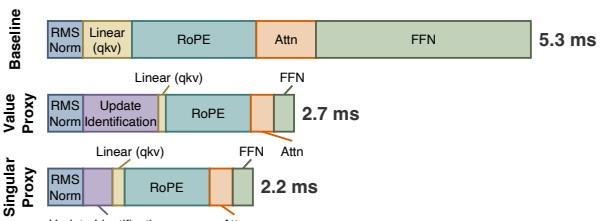

*Figure 4.* **Component-wise Latency Decomposition.** Value proxy reduces Attention and FFN overhead, but it introduces additional computational cost during update identification. Our proposed singular proxy further reduces this identification cost.

mizing the practical utility of the sparse update strategy.

## 5 Discussion

**Alternative Design Explorations.** We investigated another alternatives that ultimately proved sub-optimal. In current DLM implementations with moderate sequence lengths (*e.g.* $\leq 2048$), the Attention is relatively lightweight compared to the FFN component. We hypothesized that using the Attention output as an identifier would provide a higher-fidelity signal for update selection. However, as shown in Table 1, identification accuracy dropped significantly. Post-mortem analysis suggests this is due to the Anisotropy Masking Effect (see Section B), where the Attention output exhibits a lower signal-to-noise ratio for hidden state dynamics.

**Limitations.** While SPA-Cache demonstrates superior performance across various benchmarks, it possesses certain limitations. Like other DLM caching methods (Liu et al., 2025b; Wu et al., 2025b; Ma et al., 2025), its efficacy relies on the temporal smoothness of hidden state transitions. In high-temperature sampling regimes (*e.g.* $\tau \in [1.0, 1.5]$), the increased stochasticity can lead to reduced accuracy. Additionally, deploying SPA-Cache in distributed or tensor-parallel settings may require advanced engineering specifications, increasing the implementation complexity.

**Further study.** SPA-Cache primarily optimizes the decoding phase, leaving the prefilling latency (Time-to-First-Token, TTFT) largely unaffected. Furthermore, as current open-source DLMs do not yet support extended context windows, evaluating their performance on ultra-long sequences (*e.g.*, > 32k tokens) will be a critical area of investigation as foundation models evolve to incorporate such capabilities.

## 6 Conclusion

This paper introduces SPA-Cache, a training-free and principled caching framework designed to address the inference bottlenecks of diffusion language models. Moving beyond purely empirical observations, we provide a theoretical foundation for hidden state update identification, justifying the use of Value states as an effective proxy. By projecting the identification process into a low-dimensional manifold through our singular proxy approach, we significantly reduce overhead of update identification. Furthermore, we capitalize on the discovered layer-wise heterogeneity, utilizing an adaptive budget allocation strategy to avoid redundant computations in stable layers. Extensive benchmarks confirm that SPA-Cache delivers substantial throughput gains without compromising generation fidelity, and its seamless integration with parallel decoding underscores its versatility. We believe SPA-Cache establishes a new efficiency baseline and will catalyze further research into the practical deployment of discrete diffusion models.

# Acknowledgements

This research / project is supported by the National Research Foundation, Singapore, and Cyber Security Agency of Singapore under its National Cybersecurity R&D Programme and CyberSG R&D Cyber Research Programme Office. Any opinions, findings and conclusions or recommendations expressed in these materials are those of the author(s) and do not reflect the views of National Research Foundation, Singapore, Cyber Security Agency of Singapore as well as CyberSG R&D Programme Office, Singapore.

# Impact Statement

Generative language models carry the risk of producing biased, privacy-violating, harmful content, or offending intellectual property. Our method, designed to improve the generation efficiency of diffusion language models, may also inherit these potential negative impacts. Users and service providers should take responsibility for the generated content and strive to ensure positive social impacts.

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

# A    Proof of Theorems in Section 3

## A.1    Proof of Theorem 3.1

*Proof.* We adopt two stability assumptions:

1. **Stable Attention:** The total variation of attention weights is bounded: $\sum_j |\alpha_{ij}^{t+1} - \alpha_{ij}^t| \leq \delta_A$.

2. **Maximal Drift at $i$:** The drift of the $i$-th Value vector upper-bounds the weighted drift of the context: $\sum_j \alpha_{ij}^{t+1}\|v_j^{t+1} - v_j^t\| \leq \lambda\|v_i^{t+1} - v_i^t\|$, where $\lambda \geq 1$ is a relax constant.

We define the drift of the output vector as $\Delta h_i = h_i^{t+1} - h_i^t$. Using the triangle inequality, we separate the drift into Value shifts and Attention shifts:

$$\|\Delta h_i\| = \left\|\sum_j \alpha_{ij}^{t+1}v_j^{t+1} - \sum_j \alpha_{ij}^t v_j^t\right\| \leq \underbrace{\sum_j \alpha_{ij}^{t+1}\|v_j^{t+1} - v_j^t\|}_{\text{Weighted Value Drift } (\mathcal{D}_v)} + \underbrace{\sum_j |\alpha_{ij}^{t+1} - \alpha_{ij}^t|\|v_j^t\|}_{\text{Attention Shift } (\mathcal{E}_\alpha)}. \tag{6}$$

For the value drift term $\mathcal{D}_v$:

$$\mathcal{D}_v \leq \lambda\|v_i^{t+1} - v_i^t\|. \tag{7}$$

Let the norms be bounded such that $\|v\| \in [V_{\min}, V_{\max}]$ and $\|h\| \in [H_{\min}, H_{\max}]$ (Ruibin et al., 2020; Jacob & Ambuj, 2024). For the attention shift term $\mathcal{E}_\alpha$, using the bound $\delta_A$ and max norm $V_{\max}$:

$$\mathcal{E}_\alpha \leq \delta_A V_{\max}. \tag{8}$$

Let $\Delta_V = V_{\max} - V_{\min}$ be the maximum norm fluctuation.

$$\|v_i^{t+1} - v_i^t\| = \sqrt{(\|v_i^{t+1}\| - \|v_i^t\|)^2 + 2\|v_i^{t+1}\|\|v_i^t\|(1 - \mathcal{S}_{\cos}(v_i^t, v_i^{t+1}))} \tag{9}$$

$$\leq \sqrt{\Delta_V^2 + 2V_{\max}^2(1 - \mathcal{S}_{\cos}(v_i^t, v_i^{t+1}))} \tag{10}$$

Using the subadditivity of the square root ($\sqrt{a+b} \leq \sqrt{a} + \sqrt{b}$):

$$\|v_i^{t+1} - v_i^t\| \leq \Delta_V + V_{\max}\sqrt{2(1 - \mathcal{S}_{\cos}(v_i^t, v_i^{t+1}))}. \tag{11}$$

Substituting this back into the bound for $\|\Delta h_i\|$:

$$\|\Delta h_i\| \leq \lambda V_{\max}\sqrt{2(1 - \mathcal{S}_{v_i})} + (\lambda\Delta_V + \delta_A V_{\max}). \tag{12}$$

Let $X = \lambda V_{\max}\sqrt{2(1 - \mathcal{S}_{v_i})}$ be the signal term and $Y = \lambda\Delta_V + \delta_A V_{\max}$ be the noise term.

From the geometry of cosine similarity:

$$\mathcal{S}_{\cos}(h_i^t, h_i^{t+1}) \geq 1 - \frac{\|\Delta h_i\|^2}{2H_{\min}^2}. \tag{13}$$

Rearranging for dissimilarity $(1 - \mathcal{S})$:

$$1 - \mathcal{S}_{\cos}(h_i^t, h_i^{t+1}) \leq \frac{(X+Y)^2}{2H_{\min}^2} \leq \frac{X^2 + 2XY + Y^2}{2H_{\min}^2}. \tag{14}$$

Substituting $X^2 = 2\lambda^2 V_{\max}^2(1 - \mathcal{S}_{v_i})$:

$$1 - \mathcal{S}_{\cos}(h_i^t, h_i^{t+1}) \leq \underbrace{\frac{\lambda^2 V_{\max}^2}{H_{\min}^2}}_{C}(1 - \mathcal{S}_{\cos}(v_i^t, v_i^{t+1})) + \underbrace{\frac{2XY + Y^2}{2H_{\min}^2}}_{\epsilon}. \tag{15}$$

This concludes the proof. The output dissimilarity is linearly bounded by the input dissimilarity of token $i$, scaled by the squared condition number $C = (\lambda V_{\max}/H_{\min})^2$.    □

## A.2 Proof of Theorem 3.2

*Proof.* A standard Transformer MLP consists of RMSNorm operations, several linear projections, activation functions $\sigma(\cdot)$. Each of these components is Lipschitz continuous (Henry et al., 2018):

- Linear transformations ($\boldsymbol{W}$) are Lipschitz with constant equal to their spectral norm $\|\boldsymbol{W}\|$.
- Standard activations (ReLU, GELU, SiLU) are Lipschitz continuous with constant $L_{\text{act}} \approx 1$.
- RMSNorm is Lipschitz continuous over a domain of inputs bounded away from zero.

The composition of Lipschitz functions is Lipschitz. Thus, there exists a constant $L$ such that for any inputs $\boldsymbol{h}_1, \boldsymbol{h}_2$:

$$\|f_{\text{MLP}}(\boldsymbol{h}_1) - f_{\text{MLP}}(\boldsymbol{h}_2)\| \leq L\|\boldsymbol{h}_1 - \boldsymbol{h}_2\|. \tag{16}$$

Consider the squared Euclidean distance between the inputs $\boldsymbol{h}_1$ and $\boldsymbol{h}_2$:

$$\|\boldsymbol{h}_1 - \boldsymbol{h}_2\|^2 = \|\boldsymbol{h}_1\|^2 + \|\boldsymbol{h}_2\|^2 - 2\|\boldsymbol{h}_1\|\|\boldsymbol{h}_2\|\mathcal{S}_{\cos}(\boldsymbol{h}_1, \boldsymbol{h}_2). \tag{17}$$

Assuming the hidden states within the Transformer operate within a bounded norm range (Ruibin et al., 2020; Jacob & Ambuj, 2024), let $H_{\min} \leq \|\boldsymbol{h}.\| \leq H_{\max}$. The Euclidean distance can be exactly related to cosine similarity via:

$$\|\boldsymbol{h}_1 - \boldsymbol{h}_2\|^2 = (\|\boldsymbol{h}_1\| - \|\boldsymbol{h}_2\|)^2 + 2\|\boldsymbol{h}_1\|\|\boldsymbol{h}_2\|(1 - \mathcal{S}_{\cos}(\boldsymbol{h}_1, \boldsymbol{h}_2)). \tag{18}$$

Bounding the magnitude difference by $\Delta = H_{\max} - H_{\min}$ and the product by $H_{\max}^2$, we obtain a strict upper bound:

$$\|\boldsymbol{h}_1 - \boldsymbol{h}_2\|^2 \leq \Delta^2 + 2H_{\max}^2(1 - \mathcal{S}_{\cos}(\boldsymbol{h}_1, \boldsymbol{h}_2)). \tag{19}$$

Using the subadditivity of the square root ($\sqrt{a+b} \leq \sqrt{a} + \sqrt{b}$):

$$\|\boldsymbol{h}_1 - \boldsymbol{h}_2\| \leq \Delta + \sqrt{2}H_{\max} \cdot \sqrt{1 - \mathcal{S}_{\cos}(\boldsymbol{h}_1, \boldsymbol{h}_2)}. \tag{20}$$

Substituting Equation (20) into the Lipschitz condition (Equation (16)), we set $C = L \cdot \sqrt{2}H_{\max}$ and $\epsilon = L \cdot \Delta$:

$$\|f_{\text{MLP}}(\boldsymbol{h}_1) - f_{\text{MLP}}(\boldsymbol{h}_2)\| \leq \epsilon + C \cdot \sqrt{1 - \mathcal{S}_{\cos}(\boldsymbol{h}_1, \boldsymbol{h}_2)}. \tag{21}$$

$\square$

## A.3 Proof of Theorem 3.4

### A.3.1 PROPOSITIONS AND LEMMA FOR THE PROOF

**Proposition A.1.** *(Generalized Bernoulli Theorem) For $x \geq 0$, $1/\sqrt{1+x} \geq 1 - x/2$.*

*Proof.* The proof is straightforward by Taylor expansion of $1/\sqrt{1+x}$ at $x = 0$:

$$\frac{1}{\sqrt{1+x}} = 1 - \frac{1}{2}x + \underbrace{\frac{3}{8}x^2 - \frac{5}{16}x^3 + \ldots}_{\geq 0} \geq 1 - \frac{1}{2}x. \tag{22}$$

$\square$

**Proposition A.2.** *For any $\boldsymbol{h} \in \mathbb{R}^d$ and the $r$-rank approximation $\boldsymbol{U}_r\Lambda_r\boldsymbol{V}_r^\top$ of $\boldsymbol{W}$, the following inequality holds:*

$$\frac{\|\boldsymbol{W}\boldsymbol{h} - \boldsymbol{U}_r\Lambda_r\boldsymbol{V}_r^\top\boldsymbol{h}\|}{\|\Lambda_r\boldsymbol{V}_r^\top\boldsymbol{h}\|} \leq \frac{\lambda_{r+1}}{\lambda_r}\frac{\|\boldsymbol{h}\|}{\|\boldsymbol{V}_r^\top\boldsymbol{h}\|}. \tag{23}$$

*Proof.* For the numerator, we have

$$\|\boldsymbol{W}\boldsymbol{h} - \boldsymbol{U}_r\Lambda_r\boldsymbol{V}_r^\top\boldsymbol{h}\| \leq \|\boldsymbol{W} - \boldsymbol{U}_r\Lambda_r\boldsymbol{V}_r^\top\|\|\boldsymbol{h}\| = \lambda_{r+1}\|\boldsymbol{h}\|. \tag{24}$$

For the denominator, we have

$$\|\Lambda_r\boldsymbol{V}_r^\top\boldsymbol{h}\| \geq \lambda_r\|\boldsymbol{V}_r^\top\boldsymbol{h}\|. \tag{25}$$

Combining these two inequalities,

$$\frac{\|\boldsymbol{W}\boldsymbol{h} - \boldsymbol{U}_r\Lambda_r\boldsymbol{V}_r^\top\boldsymbol{h}\|}{\|\Lambda_r\boldsymbol{V}_r^\top\boldsymbol{h}\|} \leq \frac{\lambda_{r+1}}{\lambda_r}\frac{\|\boldsymbol{h}\|}{\|\boldsymbol{V}_r^\top\boldsymbol{h}\|}. \tag{26}$$

$\square$

**Lemma A.3.** *Let $\boldsymbol{W}_r = \Lambda_r\boldsymbol{V}_r^\top$, $\boldsymbol{o} = \boldsymbol{W}\boldsymbol{h}$, and $\boldsymbol{o}' = \boldsymbol{W}_r\boldsymbol{h}$. Then the divergence of the cosine similarity between the full and low-dimensional projections can be bounded as follows:*

$$|\mathcal{S}_{\cos}(\boldsymbol{o}_1, \boldsymbol{o}_2) - \mathcal{S}_{\cos}(\boldsymbol{o}_1', \boldsymbol{o}_2')| \leq \frac{1}{2}\left(\frac{\lambda_{r+1}}{\lambda_r}\right)^2\left(\frac{\|\boldsymbol{h}_1\|}{\|\boldsymbol{V}_r^\top\boldsymbol{h}_1\|} + \frac{\|\boldsymbol{h}_2\|}{\|\boldsymbol{V}_r^\top\boldsymbol{h}_2\|}\right)^2. \tag{27}$$

*Proof.* We start with residuals of the low-rank approximation $\boldsymbol{R} := \boldsymbol{W} - \boldsymbol{U}_r\Lambda_r\boldsymbol{V}_r^\top$. By the definition of spectral norm, we have $\|\boldsymbol{R}\| = \lambda_{r+1}$. And $\boldsymbol{R}$ and $\boldsymbol{U}_r$ are orthogonal (mathematically, $\boldsymbol{R}^\top\boldsymbol{U}_r = 0$) because the residual $\boldsymbol{R}$ lives in the span of the discarded singular vectors $\text{span}(\boldsymbol{U}) \setminus \text{span}(\boldsymbol{U}_r)$.

Denote $\boldsymbol{A} := \boldsymbol{W} - \boldsymbol{R} = \boldsymbol{U}_r\Lambda_r\boldsymbol{V}_r^\top = \boldsymbol{U}_r\boldsymbol{W}_r$. As the orthogonal matrix $\boldsymbol{U}_r$ preserves the inner product and norm, $\mathcal{S}_{\cos}(\boldsymbol{o}_1', \boldsymbol{o}_2') = \mathcal{S}_{\cos}(\boldsymbol{W}_r\boldsymbol{h}_1, \boldsymbol{W}_r\boldsymbol{h}_2) = \mathcal{S}_{\cos}(\boldsymbol{A}\boldsymbol{h}_1, \boldsymbol{A}\boldsymbol{h}_2)$, we use $\boldsymbol{A}$ rather than $\boldsymbol{W}_r$ in the following proof for clarity.

Since $\boldsymbol{R}$ and $\boldsymbol{A}$ are orthogonal: $\boldsymbol{R}^\top\boldsymbol{A} = \boldsymbol{R}^\top\boldsymbol{U}_r\Lambda_r\boldsymbol{V}_r^\top = 0 \cdot \Lambda_r\boldsymbol{V}_r^\top = 0$, the norm of $\boldsymbol{W}\boldsymbol{h}_1$ can be decomposed as follows:

$$\|\boldsymbol{o}_1\|^2 = \|\boldsymbol{W}\boldsymbol{h}_1\|^2 = \|\boldsymbol{A}\boldsymbol{h}_1 + \boldsymbol{R}\boldsymbol{h}_1\|^2 = \|\boldsymbol{A}\boldsymbol{h}_1\|^2 + \|\boldsymbol{R}\boldsymbol{h}_1\|^2 + \underbrace{2\langle\boldsymbol{A}\boldsymbol{h}_1, \boldsymbol{R}\boldsymbol{h}_1\rangle}_{=0 \text{ as } \boldsymbol{R}^\top\boldsymbol{A}=0}. \tag{28}$$

Similarly, the inner products between the adjacent features can be decomposed as follows:

$$\begin{aligned}
\langle\boldsymbol{o}_1, \boldsymbol{o}_2\rangle &= \langle\boldsymbol{W}\boldsymbol{h}_1, \boldsymbol{W}\boldsymbol{h}_2\rangle = \langle(\boldsymbol{A}+\boldsymbol{R})\boldsymbol{h}_1, (\boldsymbol{A}+\boldsymbol{R})\boldsymbol{h}_2\rangle \\
&= \langle\boldsymbol{A}\boldsymbol{h}_1, \boldsymbol{A}\boldsymbol{h}_2\rangle + \langle\boldsymbol{R}\boldsymbol{h}_1, \boldsymbol{R}\boldsymbol{h}_2\rangle + \underbrace{\langle\boldsymbol{A}\boldsymbol{h}_1, \boldsymbol{R}\boldsymbol{h}_2\rangle + \langle\boldsymbol{R}\boldsymbol{h}_1, \boldsymbol{A}\boldsymbol{h}_2\rangle}_{=0 \text{ as } \boldsymbol{R}^\top\boldsymbol{A}=0}.
\end{aligned} \tag{29}$$

Denote $a_1 := \|\boldsymbol{A}\boldsymbol{h}_1\|$, $h_1 := \|\boldsymbol{W}\boldsymbol{h}_1\|$, and $r_1 := \|\boldsymbol{R}\boldsymbol{h}_1\|$ for brevity,

$$\begin{aligned}
\mathcal{S}_{\cos}(\boldsymbol{o}_1, \boldsymbol{o}_2) - \mathcal{S}_{\cos}(\boldsymbol{o}_1', \boldsymbol{o}_2') &= \frac{\langle\boldsymbol{W}\boldsymbol{h}_1, \boldsymbol{W}\boldsymbol{h}_2\rangle}{\|\boldsymbol{W}\boldsymbol{h}_1\|\|\boldsymbol{W}\boldsymbol{h}_2\|} - \frac{\langle\boldsymbol{A}\boldsymbol{h}_1, \boldsymbol{A}\boldsymbol{h}_2\rangle}{\|\boldsymbol{A}\boldsymbol{h}_1\|\|\boldsymbol{A}\boldsymbol{h}_2\|} \\
&= \frac{\langle\boldsymbol{A}\boldsymbol{h}_1, \boldsymbol{A}\boldsymbol{h}_2\rangle + \langle\boldsymbol{R}\boldsymbol{h}_1, \boldsymbol{R}\boldsymbol{h}_2\rangle}{\|\boldsymbol{A}\boldsymbol{h}_1\|\|\boldsymbol{A}\boldsymbol{h}_2\|} \cdot \frac{\|\boldsymbol{A}\boldsymbol{h}_1\|\|\boldsymbol{A}\boldsymbol{h}_2\|}{\|\boldsymbol{W}\boldsymbol{h}_1\|\|\boldsymbol{W}\boldsymbol{h}_2\|} - \frac{\langle\boldsymbol{A}\boldsymbol{h}_1, \boldsymbol{A}\boldsymbol{h}_2\rangle}{\|\boldsymbol{A}\boldsymbol{h}_1\|\|\boldsymbol{A}\boldsymbol{h}_2\|} \quad \text{\color{teal}[by Equation (29)]} \\
&= \underbrace{\frac{\langle\boldsymbol{A}\boldsymbol{h}_1, \boldsymbol{A}\boldsymbol{h}_2\rangle}{\|\boldsymbol{A}\boldsymbol{h}_1\|\|\boldsymbol{A}\boldsymbol{h}_2\|}}_{=\mathcal{S}_{\cos}(\boldsymbol{o}_1', \boldsymbol{o}_2')\in[-1,1]}\underbrace{\left(\frac{\|\boldsymbol{A}\boldsymbol{h}_1\|\|\boldsymbol{A}\boldsymbol{h}_2\|}{\|\boldsymbol{W}\boldsymbol{h}_1\|\|\boldsymbol{W}\boldsymbol{h}_2\|} - 1\right)}_{=\frac{a_1 a_2}{h_1 h_2}-1} + \frac{\langle\boldsymbol{R}\boldsymbol{h}_1, \boldsymbol{R}\boldsymbol{h}_2\rangle}{\|\boldsymbol{W}\boldsymbol{h}_1\|\|\boldsymbol{W}\boldsymbol{h}_2\|}.
\end{aligned} \tag{30}$$

For the absolute value of the first term, we have

$$|\mathcal{S}_{\cos}(\boldsymbol{o}_1', \boldsymbol{o}_2')|\left|\frac{a_1 a_2}{h_1 h_2} - 1\right| \leq \frac{1}{2}\left(\frac{r_1}{a_1}\right)^2 + \frac{1}{2}\left(\frac{r_2}{a_2}\right)^2. \tag{31}$$

Here the inequality is derived from Proposition A.1 by appling $x = \left(\frac{r_1}{a_1}\right)^2$:

$$\frac{a_1}{h_1} = \frac{1}{\sqrt{1 + \left(\frac{r_1}{a_1}\right)^2}} \geq 1 - \frac{1}{2}\left(\frac{r_1}{a_1}\right)^2 \text{ and similarly, } \frac{a_2}{h_2} \geq 1 - \frac{1}{2}\left(\frac{r_2}{a_2}\right)^2, \tag{32}$$

$$\frac{a_1 a_2}{h_1 h_2} \geq 1 - \frac{1}{2}\left(\frac{r_1}{a_1}\right)^2 - \frac{1}{2}\left(\frac{r_2}{a_2}\right)^2 + \frac{1}{4}\left(\frac{r_1 r_2}{a_1 a_2}\right)^2 \geq 1 - \frac{1}{2}\left(\frac{r_1}{a_1}\right)^2 - \frac{1}{2}\left(\frac{r_2}{a_2}\right)^2. \tag{33}$$

So,

$$\left|\frac{a_1 a_2}{h_1 h_2} - 1\right| = 1 - \frac{a_1 a_2}{h_1 h_2} \leq \frac{1}{2}\left(\frac{r_1}{a_1}\right)^2 + \frac{1}{2}\left(\frac{r_2}{a_2}\right)^2. \tag{34}$$

For the absolute value of the second term, we can use the Cauchy-Schwarz inequality:

$$\left|\frac{\langle \boldsymbol{R}\boldsymbol{h}_1, \boldsymbol{R}\boldsymbol{h}_2\rangle}{\|\boldsymbol{W}\boldsymbol{h}_1\|\|\boldsymbol{W}\boldsymbol{h}_2\|}\right| \leq \frac{\|\boldsymbol{R}\boldsymbol{h}_1\|\|\boldsymbol{R}\boldsymbol{h}_2\|}{\|\boldsymbol{W}\boldsymbol{h}_1\|\|\boldsymbol{W}\boldsymbol{h}_2\|} = \frac{r_1 r_2}{h_1 h_2} \leq \frac{r_1 r_2}{a_1 a_2} \tag{35}$$

Combining Equations (31), (35) and (30), we have

$$|\mathcal{S}_{\cos}(\boldsymbol{o}_1, \boldsymbol{o}_2) - \mathcal{S}_{\cos}(\boldsymbol{o}_1', \boldsymbol{o}_2')| \leq \frac{1}{2}\left(\frac{r_1}{a_1}\right)^2 + \frac{1}{2}\left(\frac{r_2}{a_2}\right)^2 + \frac{r_1 r_2}{a_1 a_2} = \frac{1}{2}\left(\frac{r_1}{a_1} + \frac{r_2}{a_2}\right)^2. \tag{36}$$

By Proposition A.2,

$$\frac{r_1}{a_1} \leq \frac{\lambda_{r+1}}{\lambda_r}\frac{\|\boldsymbol{h}_1\|}{\|\boldsymbol{V}_r^\top \boldsymbol{h}_1\|}, \text{ and } \frac{r_2}{a_2} \leq \frac{\lambda_{r+1}}{\lambda_r}\frac{\|\boldsymbol{h}_2\|}{\|\boldsymbol{V}_r^\top \boldsymbol{h}_2\|}. \tag{37}$$

Substituting Equation (37) into Equation (36), the final bound is

$$|\mathcal{S}_{\cos}(\boldsymbol{o}_1, \boldsymbol{o}_2) - \mathcal{S}_{\cos}(\boldsymbol{o}_1', \boldsymbol{o}_2')| \leq \frac{1}{2}\left(\frac{\lambda_{r+1}}{\lambda_r}\right)^2\left(\frac{\|\boldsymbol{h}_1\|}{\|\boldsymbol{V}_r^\top \boldsymbol{h}_1\|} + \frac{\|\boldsymbol{h}_2\|}{\|\boldsymbol{V}_r^\top \boldsymbol{h}_2\|}\right)^2. \tag{38}$$

$\square$

### A.3.2 PROOF OF THEOREM 3.4

*Proof.* With Lemma A.3, it suffices to show that for any $\boldsymbol{h}_1, \boldsymbol{h}_2 \in \text{span}(\boldsymbol{V}_r)$, we have $\|\boldsymbol{h}_1\| = \|\boldsymbol{V}_r^\top \boldsymbol{h}_1\|$ and $\|\boldsymbol{h}_2\| = \|\boldsymbol{V}_r^\top \boldsymbol{h}_2\|$, which establishes the conclusion of Theorem 3.4. This is straightforward because $\boldsymbol{V}_r$ is an orthonormal basis of $\text{span}(\boldsymbol{V}_r)$, so for any $\boldsymbol{h} \in \text{span}(\boldsymbol{V}_r)$, we have $\|\boldsymbol{h}\| = \|\boldsymbol{V}_r^\top \boldsymbol{h}\|$. $\square$

*Remark* A.4. (**Connection with MaxLogits**) It is widely observed that the spectral norm of the attention projection matrix, $\|\boldsymbol{W}\|_2 = \lambda_1$, tends to become large during the training of large language models. While this behavior can lead to numerical instability (Steven et al., 2022), it also incidentally benifits the error bound of our proposed singular proxy identifier.

## B Failure Analysis

As discussed in Section 3.2, using the attention output as the update identifier has the worst accuracy among all configurations in Table 1. Intuitively, the attention output should be eligbile. We then dive deeper to investigate its failure mechanism. The core issue lies in that the anisotropy of the attention outputs have been masked, reducing the sensitivity of the similarity metric to subtle semantic drifts.

Specifically, the anisotropy masks individual token characteristics, thereby reducing the sensitivity of the similarity metric to subtle semantic drifts (Ethayarajh, 2019; Gao et al., 2019; Timkey & van Schijndel, 2021; Zhang et al., 2025c; Fang et al., 2025). It is widely observed that hidden states in Transformers, specifically the Value states $\boldsymbol{v}_i$ in our context, can be decomposed into a common vector $\boldsymbol{c}$ and an independent semantic signal $\boldsymbol{s}_i$:

$$\boldsymbol{v}_i = \boldsymbol{c} + \boldsymbol{s}_i, \quad \text{where} \quad \mathbb{E}[\boldsymbol{s}_i] = \boldsymbol{0} \text{ and } \|\boldsymbol{c}\| > 0. \tag{39}$$

While this phenomenon is present, the common vector $\boldsymbol{c}$ remains relatively non-dominant ($\|\boldsymbol{s}_i\| > \|\boldsymbol{c}\|$), allowing the states to maintain sufficient orthogonality ($\mathbb{E}_{i \neq j}[\mathcal{S}_{\cos}(\boldsymbol{v}_i, \boldsymbol{v}_j)] \approx 0$). This preservation of semantic variance is visualized in

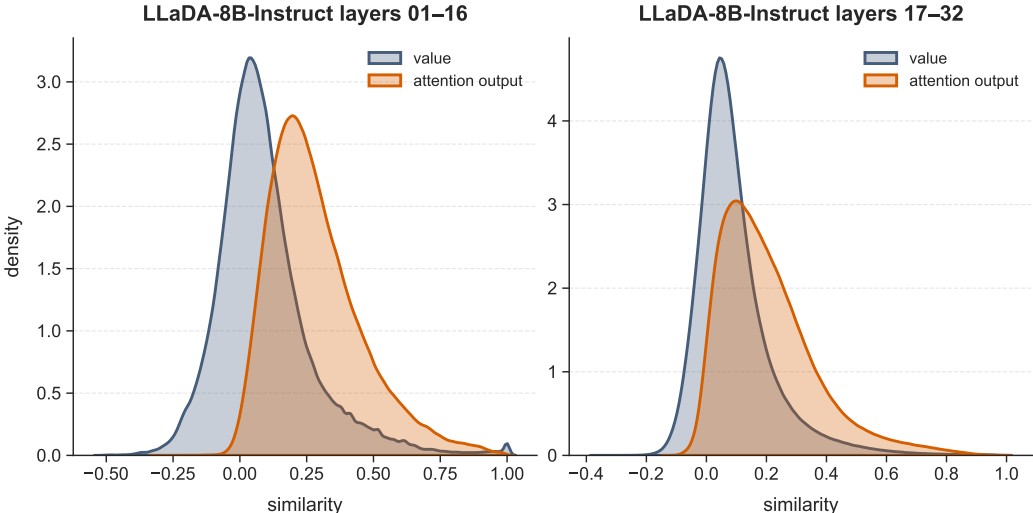

*Figure 5.* **Empirical Evidence of Anisotropy Problem in Attention Outputs.** The density plots compare the cosine similarity distributions of value states (blue) versus attention outputs (orange). While value states exhibit an isotropic distribution centered near zero (orthogonality), the attention outputs show a significant positive shift in mean similarity.

*Table 6.* Hyperparameters for fitted piecewise Gaussian function.

| MODEL | $l_p$ | $\rho_p$ | $\rho_1$ | $\rho_L$ |
|---|---|---|---|---|
| LLADA-8B-INSTRUCT | 24 | 25% | 3% | 13% |
| LLADA-1.5 | 25 | 25% | 3% | 13% |
| DREAM-V0-INSTRUCT-7B | 14 | 30% | 5% | 25% |

Figure 5. However, this representation collapses when states are aggregated into the attention output $\boldsymbol{h}_i = \sum_j \alpha_{ij}\boldsymbol{v}_j$. Given the stochastic constraint $\sum_j \alpha_{ij} = 1$, the transformation is expressed as:

$$\boldsymbol{h}_i = \sum_j \alpha_{ij}(\boldsymbol{c} + \boldsymbol{s}_j) = \boldsymbol{c} + \sum_j \alpha_{ij}\boldsymbol{s}_j. \tag{40}$$

Here, the attention output $\boldsymbol{h}_i$ becomes heavily dominated by the common direction $\boldsymbol{c}$. Because the semantic signals $\boldsymbol{s}_j$ are independent and zero-mean, their weighted sum undergoes a form of "signal cancellation," leading to $\|\boldsymbol{c}\| \gg \|\sum_j \alpha_{ij}\boldsymbol{s}_j\|$. Consequently, the similarity distribution $\mathcal{S}_{\cos}(\boldsymbol{h}_i, \boldsymbol{h}_j)$ shifts toward a narrow cone where cosine similarities approach 1. This anisotropy renders the identifier insensitive to the subtle semantic drifts required for accurate update tracking.

## C   Details on Adaptive Budget Allocation

In Section 3.4, we characterized the layer-wise heterogeneity of hidden state dynamics for LLaDA-8B (Nie et al., 2025). In this section, we extend our analysis to a broader range of models. Specifically, the distribution profiles for LLaDA-1.5B (Zhu et al., 2025) and Dream-7B (Ye et al., 2025) are illustrated in Figure 6.

While these models exhibit a general trend consistent with LLaDA-8B, the parameters for the fitted piecewise Gaussian distributions vary across architectures. We provide the specific hyperparameters employed for these models in Table 6. It is important to emphasize that these hyperparameters remain fixed across all evaluated benchmarks, demonstrating the robust generalization capabilities of our proposed method.

## D   Experimental Details

In our experiments, we follow the decoding hyperparameters established by Liu et al. (2025b). For completeness, these detailed configurations are provided in Table 7.

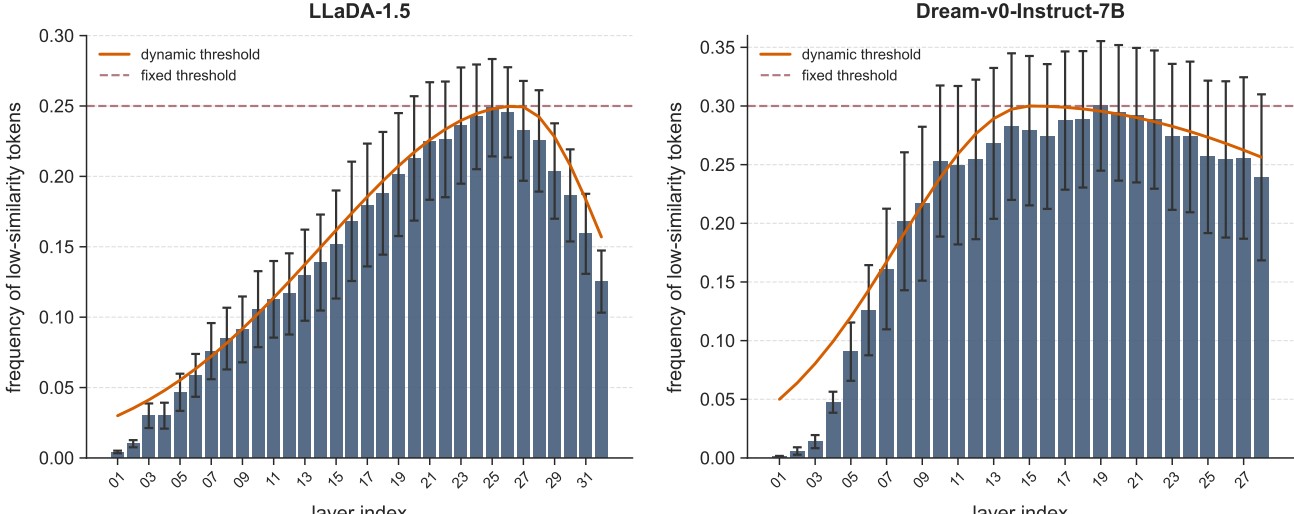

*Figure 6.* Distribution of drift for LLaDA-1.5 (Zhu et al., 2025) and Dream-v0-Instruct-7B (Ye et al., 2025).

*Table 7.* Experiment settings across benchmarks.

| TASK | LLADA-8B-INSTRUCT & LLADA-1.5 | | | DREAM-V0-INSTRUCT-7B | | |
|---|---|---|---|---|---|---|
| | N-SHOT | GEN. LENGTH | BLOCK LENGTH | N-SHOT | GEN. LENGTH | BLOCK LENGTH |
| GSM8K | 4 | 256 | 8 | 8 | 256 | 256 |
| GPQA | 5 | 128 | 64 | 5 | 256 | 256 |
| MATH500 | 4 | 256 | 32 | 4 | 256 | 256 |
| BBH | 3 | 256 | 256 | 3 | 256 | 256 |
| MMLU-PRO | 5 | 256 | 256 | 5 | 256 | 256 |
| MBPP | 3 | 512 | 32 | 3 | 256 | 256 |
| HUMANEVAL | 0 | 512 | 32 | 0 | 256 | 256 |

For the Fast-dLLM baseline (Wu et al., 2025b), which requires the block size to be smaller than the generation length to achieve effective acceleration, we set the block size to 32 uniformly across all models and benchmarks. Other method-specific hyperparameters, such as the refresh interval in dLLM-Cache (Liu et al., 2025b), are kept consistent with their original configurations.

Regarding batch sizes, we use a batch size of 16 for all LLaDA-8B tasks. For the Dream-7B model, the batch size is set to 4. This specific reduction was necessitated by Out-of-Memory (OOM) issues encountered with Fast-dLLM during long-prompt benchmarks (*e.g.* MMLU-pro). While other evaluated methods could support larger batch sizes for higher throughput, we maintain a consistent batch size of 4 for Dream-7B to ensure a fair comparison across all baselines.

Regarding the selection of the proxy rank $r$, we set $r = 128$ for the LLaDA models (Nie et al., 2025; Zhu et al., 2025), where the hidden dimension is $d = 4096$. This choice is consistent with our ablation studies detailed in Section 4.4. For the Dream model (Ye et al., 2025), which utilizes Grouped-Query Attention (GQA) (Ainslie et al., 2023), the Value projection dimension is significantly smaller ($d = 512$). Consequently, we adopt a more aggressive proxy rank of $r = 32$ to maintain a comparable acceleration.

## E  Additional Experiment Results

We further evaluate our approach on LLaDA-1.5B (Zhu et al., 2025), with results detailed in Table 8. The performance trends closely mirror those observed for the larger LLaDA-8B and Dream-7B models (Table 2), underscoring the architectural robustness of our method. Notably, SPA-Cache consistently establishes a new state-of-the-art across all benchmarks, delivering substantial throughput gains while preserving the original model's generation quality with negligible degradation.

We also compare with dKV-Cache (Ma et al., 2025), d2Cache (Jiang et al., 2025), and ElasticCache (Nguyen-Tri et al., 2025)

*Table 8.* Comparision of different caching methods on LLaDA-1.5 (Zhu et al., 2025).

| TASK | METHOD | TPS ↑ | TTFT (MS) ↓ | ACCURACY ↑ | MEM. (GB) ↓ |
|---|---|---|---|---|---|
| MATHEMATICS & SCIENCE | | | | | |
| GSM8K | BASELINE | 28.9 (×1.0) | 28.04 | 81.96 (±1.1) | 62.3 |
| | + DLLM-CACHE | 68.5 (×2.4) | 28.03 | 81.43 (±1.1) | 63.8 |
| | + FAST-DLLM | 58.0 (×2.0) | 28.09 | 82.18 (±1.1) | 64.1 |
| | + OURS | 189.9 (×6.6) | 31.13 | 81.50 (±1.1) | 57.5 |
| GPQA | BASELINE | 21.0 (×1.0) | 44.90 | 27.46 (±2.1) | 102.4 |
| | + DLLM-CACHE | 29.2 (×1.4) | 44.72 | 27.90 (±2.1) | 77.0 |
| | + FAST-DLLM | 39.4 (×1.9) | 44.26 | 27.01 (±2.1) | 91.4 |
| | + OURS | 49.2 (×2.3) | 47.68 | 27.90 (±2.1) | 96.6 |
| MATH500 | BASELINE | 35.5 (×1.0) | 23.23 | 34.40 (±0.6) | 63.4 |
| | + DLLM-CACHE | 74.2 (×2.1) | 23.28 | 33.92 (±0.6) | 57.9 |
| | + FAST-DLLM | 83.9 (×2.4) | 23.19 | 33.54 (±0.6) | 66.5 |
| | + OURS | 201.8 (×5.7) | 23.13 | 34.56 (±0.6) | 51.6 |
| GENERAL QA | | | | | |
| BBH | BASELINE | 22.3 (×1.0) | 16.22 | 56.61 (±0.6) | 37.0 |
| | + DLLM-CACHE | 36.0 (×1.6) | 16.35 | 57.00 (±0.6) | 45.0 |
| | + FAST-DLLM | 103.9 (×4.7) | 16.22 | 54.66 (±0.6) | 36.3 |
| | + OURS | 131.5 (×5.9) | 16.27 | 57.43 (±0.6) | 34.3 |
| MMLU-PRO | BASELINE | 21.4 (×1.0) | 43.99 | 38.51 (±0.4) | 95.3 |
| | + DLLM-CACHE | 51.7 (×2.4) | 42.12 | 38.30 (±0.4) | 87.5 |
| | + FAST-DLLM | 84.3 (×4.0) | 43.41 | 38.31 (±0.4) | 79.3 |
| | + OURS | 233.2 (×10.9) | 43.90 | 38.36 (±0.4) | 79.1 |
| CODE | | | | | |
| MBPP | BASELINE | 6.7 (×1.0) | 28.61 | 40.80 | 70.7 |
| | + DLLM-CACHE | 9.0 (×1.3) | 28.22 | 40.80 | 74.2 |
| | + FAST-DLLM | 13.3 (×2.0) | 28.25 | 37.20 | 74.1 |
| | + OURS | 69.4 (×10.4) | 29.44 | 40.00 | 69.5 |
| HUMANEVAL | BASELINE | 46.8 (×1.0) | 14.30 | 45.83 | 51.2 |
| | + DLLM-CACHE | 43.1 (×0.9) | 14.28 | 38.41 | 55.7 |
| | + FAST-DLLM | 92.8 (×2.0) | 14.23 | 41.71 | 50.3 |
| | + OURS | 143.7 (×3.1) | 14.23 | 46.88 | 48.9 |

in Table 9. Our method outperforms these baselines in throughput (TPS) while maintaining accuracy. The performance gap stems from architectural compatibility: existing methods often rely on explicit attention weight computation, which is incompatible with optimized kernels like FlashAttention. In contrast, our method detects shifts using a singular proxy, which maintains compatibility with FlashAttention and achieves higher acceleration at the same or higher accuracy.

# F  Additional Observations

Figure 7 demonstrates that the Value Proxy accurately forecasts FFN output drifts. The near-identical behavior of our proposed singular proxy to the Value Proxy empirically validates Theorem 3.4, confirming that a low-rank approximation effectively captures value-space drift.

*Table 9.* Comparison against dKV-Cache (Ma et al., 2025), Elastic-Cache (Nguyen-Tri et al., 2025), and d2Cache (Jiang et al., 2025) on LLaDA-8B-Instruct (Nie et al., 2025) and Dream-v0-Instruct-7B (Ye et al., 2025).

| TASK | METHOD | LLaDA-8B-INSTRUCT | | | DREAM-V0-INSTRUCT-7B | | |
|---|---|---|---|---|---|---|---|
| | | TPS ↑ | TTFT (MS) ↓ | ACCURACY ↑ | TPS ↑ | TTFT (MS) ↓ | ACCURACY ↑ |
| GSM8K | VANILLA | 29.67 $_{(1.0\times)}$ | 28.0 | 78.62 $_{(\pm1.13)}$ | 17.86 $_{(1.0\times)}$ | 23.6 | 75.21 $_{(\pm1.19)}$ |
| | DKV-CACHE | 56.97 $_{(1.9\times)}$ | 28.9 | 78.01 $_{(\pm1.14)}$ | 22.17 $_{(1.2\times)}$ | 24.4 | 74.82 $_{(\pm1.20)}$ |
| | ELASTIC-CACHE | 48.00 $_{(1.6\times)}$ | 27.4 | 77.41 $_{(\pm1.15)}$ | 35.77 $_{(2.0\times)}$ | 23.9 | 74.86 $_{(\pm1.22)}$ |
| | D2CACHE | 26.85 $_{(0.9\times)}$ | 28.7 | 76.42 $_{(\pm1.17)}$ | 28.92 $_{(1.6\times)}$ | 24.9 | 78.70 $_{(\pm1.13)}$ |
| | OURS | **190.73** $_{(\mathbf{6.4\times})}$ | 28.7 | 78.24 $_{(\pm1.14)}$ | **45.97** $_{(\mathbf{2.6\times})}$ | 24.5 | 77.56 $_{(\pm1.15)}$ |
| MBPP | VANILLA | 5.75 $_{(1.0\times)}$ | 28.7 | 39.20 | 36.51 $_{(1.0\times)}$ | 22.2 | 57.40 |
| | DKV-CACHE | 7.83 $_{(1.4\times)}$ | 28.1 | 39.20 | 39.56 $_{(1.1\times)}$ | 22.2 | 56.40 |
| | ELASTIC-CACHE | 16.25 $_{(2.8\times)}$ | 29.2 | 34.80 | 85.60 $_{(2.3\times)}$ | 22.1 | 52.80 |
| | D2CACHE | 25.08 $_{(4.4\times)}$ | 28.2 | 39.60 | 41.87 $_{(1.1\times)}$ | 21.6 | 57.60 |
| | OURS | **46.12** $_{(\mathbf{8.0\times})}$ | 28.7 | 39.00 | **114.85** $_{(\mathbf{3.1\times})}$ | 22.6 | 57.60 |

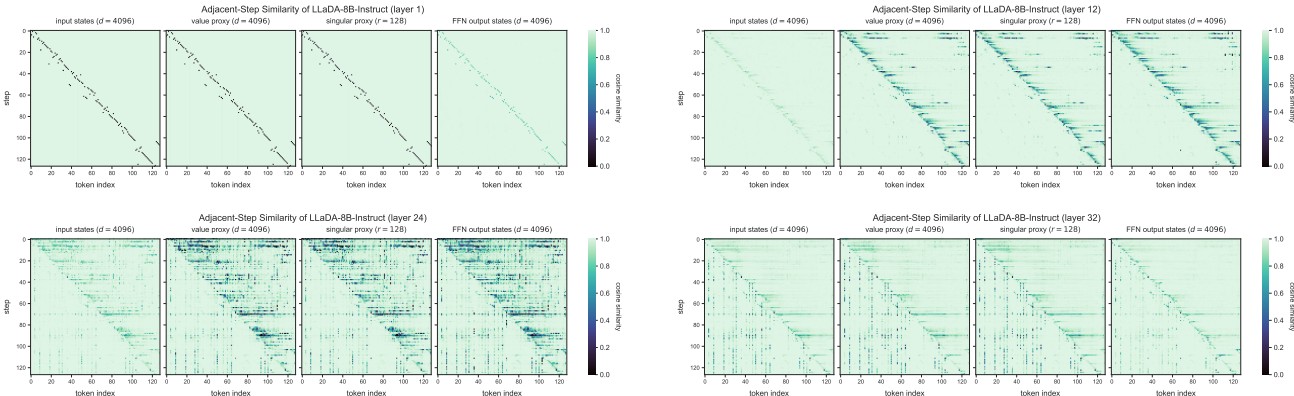

*Figure 7.* **Extended analysis of adjacent-step state similarities.** We visualize the similarities across four features (input, value, singular proxy, and output) for representative layers (1, 12, 24, and 32) of LLaDA-8B-Instruct as in Figure 1. The value proxy uncovers drift in the final FFN output more effectively than input states. Furthermore, the singular proxy's near-identical behavior to the value proxy suggests that a low-rank approximation effectively captures value-space drift.

