# OpenReview forum: "Singular Proxies for Adaptive Caching in Diffusion Language Models"
_ICML.cc/2026/Conference — ICML 2026 regular_

### Official Review · Reviewer_nV2z · 2026-02-15

**Soundness:** 3
**Presentation:** 3
**Significance:** 2
**Originality:** 2
**Overall Recommendation:** 4
**Confidence:** 4

**Summary:**

Diffusion LLMs, unlike conventional autoregressive LLMs, demonstrate the potential to dramatically accelerate inference through parallel decoding. However, from a practical perspective, diffusion LLMs require recomputation of bidirectional attention, making it difficult to utilize the KV caching used in conventional LLMs. This recomputation, in turn, leads to slower latency compared to conventional LLMs. To address this issue, caching methods for diffusion LLMs, such as Fast-dLLM, dKV-Cache, d2Cache, and dLLM-Cache, have been actively proposed recently. However, these existing methods either focus on caching with heuristic-based updates, are computationally heavy in the update-critical token identification process, or apply a uniform update budget across all layers. To address this issue, this paper proposes SPA-Cache. By introducing a singular proxy and layer-aware caching budget, SPA-Cache demonstrates superior speedup over existing methods.

**Compliance With Llm Reviewing Policy:**

Affirmed.

**Final Justification:**

The additional experimental results and clarifications on the theorems have addressed my concerns. I have decided to increase my rating from 3 to 4 as my final rating.

**Key Questions For Authors:**

Please refer to the Weaknesses section above.

**Limitations:**

Yes.

**Strengths And Weaknesses:**

**Strengths**

1. The paper tackles an important and timely challenge in KV caching for dLLMs.
2. The paper is well-written and easy to follow.
3. SPA-Cache is supported by theoretical analysis.
4. Experimental results show the superior performance of SPA-Cache.

**Weaknesses**

1. Lack of comparison with dKV-Cache and d2Cache: Although these methods are discussed in related works, they are omitted from the experimental comparison.
2. Lack of discussion regarding Elastic-Cache [1]: The paper omits Elastic-Cache, a seminal paper on the KV caching method for dLLMs, and it also uses layer-aware cache budgeting. Experimental comparison with Elastic-Cache is also necessary.
3. There is a significant difference between the TPS (164.88) of the Value row in Table 1 and the TPS (68.62) of the dLLM-Cache row in Table 2. However, if I understood correctly, there is no methodological difference. If this TPS difference comes from implementation engineering, the TPS improvement of SPA-Cache (190.73) compared to dLLM-Cache (164.88) under the same implementation seems marginal.
4. Adaptive budget relies on heuristics and requires hyperparameter tuning. It would be beneficial if there is a smarter adaptive budget allocation strategy.
5. It appears that the assumptions of Theorem 3.1 are strong, and the bounds in Theorems 3.1 and 3.2 seem loose due to the ambiguous coefficient term C. I think we can also establish similar theorems for Keys and Queries with these strong assumptions and loose bounds. Please correct me if I'm wrong.

[1] Attention Is All You Need for KV Cache in Diffusion LLMs (ICLR 2026)

---

> ### Author Rebuttal · Authors · 2026-03-31
>
> We thank Reviewer `nV2z` for acknowledging our theoretical contributions and superior experimental results. Our responses to the concerns are provided below. Additional supplementary figures and tables are available at https://anon0728.github.io/icml-2607-rebuttal/
>
> ---
>
> > W1 & W2: Lack of comparison with dKV-Cache, d2Cache, and ElasticCache
>
> We have included a comparison with dKV-Cache, d2Cache, and ElasticCache below. Our method outperforms these baselines in throughput (TPS) while maintaining accuracy. (*Other experimental results on Dream and MBPP benchmarks are provided in Suppl. Tab. 1.*)
>
>
> LLaDA: GSM8K
>
> |  | TPS | Acc |
> | --- | --- |--- |
> | Vanilla | 29.67 | 78.62 |
> | dKV-Cache | 56.97 | 78.01 |
> | Elastic-Cache | 48.00 | 77.41 |
> | d2Cache | 26.85 | 76.42 |
> | Ours | 190.73 | 78.24 |
>
>
> LLaDA: MBPP
>
> |  | TPS | Acc |
> | --- | --- |--- |
> | Vanilla | 5.75 | 39.20 |
> | dKV-Cache | 7.83 | 39.20 |
> | Elastic-Cache | 16.25 | 34.80 |
> | d2Cache | 25.08 | 39.60 |
> | Ours | 46.12 | 39.00 |
>
>
> The performance gap stems from architectural compatibility: existing methods often rely on explicit attention weight computation, which is incompatible with optimized kernels like FlashAttention. In contrast, our method detects shifts using a singular proxy, which maintains compatibility with FlashAttention and achieves higher acceleration at the same or higher accuracy.
>
> ---
>
> > W3: The significant difference between the TPS (165) of the Value row in Table 1 and the TPS (69) of the dLLM-Cache row in Table 2 comes from implementation engineering. The TPS methodological improvement of SPA-Cache (191) compared to 165 seems marginal.
>
> The discrepancy between the TPS values in Main Manuscript Tab. 1 (165) and Tab. 2 (69) for dLLM-Cache arises from our targeted optimizations. The original implementation contains numerous synchronization points, causing CPU-GPU stalls (computation bubbles) and suboptimal hardware utilization. We addressed this by re-ordering the computational graph and replacing inefficient operators. We believe these practical improvements are valuable for the community's development efforts.
>
> Furthermore, the improvement from 165 to 191 (via our adaptive singular proxy) is non-marginal in context. Compared to the vanilla implementation (30 TPS), our method further achieves additional 87% relative speedup, which is a substantial gain in high-performance inference scenarios.
>
> ---
>
> > W4: Adaptive budget relies on heuristics and requires hyperparameter tuning. It would be beneficial if there is a smarter adaptive budget allocation strategy.
>
> Hyperparameter tuning is minimal and can be automated through a one-time profiling of token drift. In practice, we profile the shift distribution using only 16 samples, a process requiring approximately two minutes. This architecture-specific profiling represents a negligible overhead and can be seamlessly integrated into automated deployment pipelines.
>
> Furthermore, our budget allocation is robust to the selection of hyperparameters. We conducted a sensitivity analysis by introducing noise to the peak layer $l_p$ (Suppl. Fig. 3). As shown in Suppl. Tab. 4, these perturbations result in only minor accuracy fluctuations (0.5-2%). These results demonstrate that our budget allocation strategy is robust to hyperparameter selection and moderate perturbations.
>
>
> ---
>
> > W5: It appears that the assumptions of Theorem 3.1 are strong, and the bounds in Theorems 3.1 and 3.2 seem loose due to the ambiguous coefficient term C. I think we can also establish similar theorems for Keys and Queries with these strong assumptions and loose bounds.
>
> Empirical Validation of Theorem 3.1 Assumptions:
>
> 1. Bounded Norms: Suppl. Fig 5 shows norms of $h$ and $v$ are strictly bounded (e.g., 10–20 for LLaDA Layer 4).
> 2. Stable Attention: Suppl. Fig 6 confirms the variation factor $\delta_A<0.3$ in most cases.
> 3. Bounded Drift: Suppl. Fig 4 shows that typically drift factor $\lambda<3$.
>
> Bound in Theorem 3.1
>
> - In Suppl. Fig 7, setting $C=1.0$ covers $>99\%$ of the dissimilarity in $h$, confirming empirical consistency.
>
> Bound in Theorem 3.2
>
> - Given $|f(h)| = |Wh| = L|h|$, where $L$ is the Lipschitz constant/spectral norm of W. As the $|h|$ is bounded, we can approximate it using $H_{\max}$. The normalized version of Appendix Eq. 21 becomes: $|\frac{f(h_1)}{LH_{\max}}-\frac{f(h_2)}{LH_{\max}}| \le \sqrt{2(1-S(h_1, h_2))}$. Thus, the bound for the normalized difference is $\sqrt{2}$ rather than an arbitrary loose term.
>
> Extension to Keys/Queries
>
> - Theorem 3.1 relies on the attention output being a linear combination of V (Appendix A.1, Eq. 6). In contrast, the relationship between Q/K and the output is intrinsically non-linear due to the softmax operation. This non-linearity precludes the direct extension of our derivation.
> - This is emperically validated by Main Manuscript Tab. 1, where utilizing Q/K fails to accurately capture drifts, resulting in accuracy degradation.

---

> > ### Author Rebuttal · Reviewer_nV2z · 2026-04-01
> >
> > Thank you for the response. The additional experimental results and clarifications on the theorems have addressed my concerns. I have decided to increase my rating from 3 to 4 as my final rating.

---

### Official Review · Reviewer_fjLA · 2026-03-07

**Soundness:** 4
**Presentation:** 2
**Significance:** 3
**Originality:** 3
**Overall Recommendation:** 5
**Confidence:** 4

**Summary:**

SPA-Cache is a training-free framework designed to overcome the inference latency in Diffusion Language Models,which often lack efficient caching mechanisms because of their non-causal structure. The authors identify two primary bottlenecks in current methods. First, the computational cost of identifying which tokens to update is often too high. Second, applying a uniform update budget across all layers is inefficient. To solve these issues, the paper introduces a singular proxy. This is a low-dimensional subspace derived from Singular Value Decomposition. It allows the model to identify critical tokens early in the process with minimal overhead. Furthermore, the authors implement an adaptive budget allocation strategy. Across seven different benchmarks, SPA-Cache achieves up to an 8x throughput improvement. It maintains high generation quality while outperforming previous state-of-the-art baselines.

**Compliance With Llm Reviewing Policy:**

Affirmed.

**Final Justification:**

The rebuttal adressed my concern.

**Key Questions For Authors:**

Quantitative Analysis of Step-wise Drift Variation:

Figure 1 qualitatively demonstrates similarity across 120 steps, yet the current budget allocation is primarily layer-wise and static across the temporal dimension. Did the authors quantitatively compare the average magnitude of drift—specifically the fraction of "drifting tokens"—across different stages of the diffusion process (e.g., early high-noise steps versus late refinement steps)? A response showing whether drift intensity remains constant or fluctuates across steps would help justify the decision to use a step-invariant budget.

**Limitations:**

yes

**Strengths And Weaknesses:**

Strengths:
1. Solid Theoretical Foundation
The paper is grounded in rigorous theory, establishing through Theorems 3.1 and 3.2 that Value state stability is a reliable proxy for the entire Transformer block's output stability. A key contribution is Theorem 3.4, which provides an analytical bound for similarity preservation, proving that the low-dimensional singular proxy is a mathematically sound approximation rather than a mere heuristic.

2. Insightful Methodology and Robust Evaluation
The core strength of the methodology lies in the insight behind the Singular Proxy. This proxy efficiently uncovers latent semantic drifts. By applying SVD to the Value projection matrix, the authors isolate a low-dimensional subspace that captures the most critical directional changes in token states. This approach recognizes that raw input states often appear uniformly stable. However, their projection through principal singular vectors reveals significant drifts that eventually manifest in the final FFN output. This mechanism reduces identification overhead significantly. It enables an 8x throughput improvement without sacrificing the precision of the updates too much.

Weakness:

While the paper is technically robust and offers a valuable contribution to the efficiency of Diffusion Language Models, the clarity of the presentation in Figure 1 could be further enhanced. Specifically, the figure illustrates that the low-dimensional singular proxy captures drifts hidden in the input states, yet it omits a direct visual comparison with the full-dimensional Value Proxy used in traditional baselines. The addition of such a comparison would provide a more intuitive validation of the Similarity Preservation property proved in Theorem 3.4. Furthermore, the figure caption lacks a detailed explanation of the relationship between the decoding steps and the diagonal drift patterns, which may make it difficult for readers to fully grasp the physical significance of these signals within the context of the model's iterative unmasking process.

---

> ### Author Rebuttal · Authors · 2026-03-31
>
> We thank the Reviewer `fjLA` for the constructive suggestions to improve the clarity of our manuscript. For the concerns and questions, here are our responses. supplementary Tables/Figures are available at [https://anon0728.github.io/icml-2607-rebuttal](https://anon0728.github.io/icml-2607-rebuttal/):
>
> ---
>
> > W: Figure 1 could be further enhanced. Specifically, the figure illustrates that the low-dimensional singular proxy captures drifts hidden in the input states, yet it omits a direct visual comparison with the full-dimensional Value Proxy used in traditional baselines. The addition of such a comparison would provide a more intuitive validation of the Similarity Preservation property proved in Theorem 3.4. Furthermore, the figure caption lacks a detailed explanation of the relationship between the decoding steps and the diagonal drift patterns, which may make it difficult for readers to fully grasp the physical significance of these signals within the context of the model's iterative unmasking process.
>
> We have provided an enhanced version of Figure 1, including additional layers and a direct comparison with the Value Proxy, in Suppl. Fig. 1.
>
> The updated visualization demonstrates that the Value Proxy accurately forecasts FFN output drifts. The near-identical behavior of our proposed singular proxy to the Value Proxy empirically validates Theorem 3.4, confirming that a low-rank approximation effectively captures value-space drift.
>
> Furthermore, we have clarified the physical significance of the diagonal patterns in the caption:
>
> - Initial Layers: Drift is primarily confined to recently decoded tokens, resulting in a sparse diagonal structure. Off-diagonal elements indicate non-strict left-to-right decoding patterns in dLLM.
> - Intermediate Layers: Drift propagates via self-attention, forming an upper-triangular pattern. This indicates that recently decoded tokens predominantly influence future positions: specifically those will be unmasked for subsequent decoding steps.
> - Final Layers: By the final layer (Layer 32), a lower-triangular pattern emerges. This suggests that certain tokens continue to drift across all steps post-decoding. Most shifts occurred in tokens proximal to the most recent output. Conversely, distant tokens, those less likely to be decoded in the immediate future, maintain stable representations.
>
> We will incorporate these detailed explanations and analyses in the next revision.
>
> ---
>
> > Q: Figure 1 qualitatively demonstrates similarity across 120 steps, yet the current budget allocation is primarily layer-wise and static across the temporal dimension. Did the authors quantitatively compare the average magnitude of drift, specifically the fraction of "drifting tokens", across different stages of the diffusion process (e.g., early high-noise steps versus late refinement steps)? A response showing whether drift intensity remains constant or fluctuates across steps would help justify the decision to use a step-invariant budget.
>
> We have included a quantitative analysis of the fraction of "drifting tokens" across layers and decoding steps in Suppl. Fig. 2.
>
> To quantify drift intensity, we plotted the fraction of drifting tokens across a range of similarity thresholds ($0.1–0.9$). Our results indicate that middle layers exhibit significantly higher drift magnitudes compared to early and late layers. This distribution aligns with our observations in Section 3.4 and further motivates our layer-wise adaptive budget allocation. Crucially, the drift magnitude remains relatively stable across the temporal (decoding step) dimension. This empirical evidence justifies our decision to employ a step-invariant budget while focusing on layer-wise adaptation.
>
> We will incorporate these detailed analyses in the next revision for more intuitive understanding.

---

> > ### Author Rebuttal · Reviewer_fjLA · 2026-04-03
> >
> > Thank you for the response. The clarifications and additional figure on the theorems have addressed my concerns.

---

### Official Review · Reviewer_CQFb · 2026-03-11

**Soundness:** 4
**Presentation:** 3
**Significance:** 4
**Originality:** 4
**Overall Recommendation:** 5
**Confidence:** 4

**Summary:**

The submission introduces SPA-Cache, a training-free caching framework designed to accelerate inference in Diffusion Language Models (DLMs). DLMs utilize bidirectional attention to decode in arbitrary orders, which traditionally precludes the use of standard autoregressive KV-caches and necessitates full recomputation at every step. SPA-Cache addresses this through two primary optimizations: (1) the derivation of a low-dimensional singular proxy for identifying "drifting" tokens that require updates, and (2) an adaptive budget allocation strategy that reduces computation in stable layers while concentrating updates on volatile ones. The authors use SVD to project Value states into a compressed subspace, significantly reducing the overhead of similarity checks. Extensive testing on LLaDA and Dream models reveals up to an 8x throughput improvement over vanilla decoding and a 2-4x speedup over existing caching baselines.

**Compliance With Llm Reviewing Policy:**

Affirmed.

**Key Questions For Authors:**

1. Theorem 3.1 relies on the assumption of "Stable Attention" (bounded total variation of weights). In early stages of diffusion (high noise), does this assumption still hold, and how does it affect the accuracy of the singular proxy?
2. The adaptive allocation uses a piecewise Gaussian function. How sensitive is the model's performance to the choice of the peak layer $l_{p}$ and the peak update ratio $\rho_{p}$ across different architectures?
3. Table 1 shows "Attn. Input" is faster but "Value" is more accurate. Can the authors discuss the trade-off between these two identifier choices in more detail?

**Limitations:**

The authors acknowledge that the efficacy of the method depends on the temporal smoothness of hidden state transitions and may struggle in high-temperature sampling regimes. Additionally, implementation complexity in distributed settings remains an area for further investigation.

**Strengths And Weaknesses:**

Soundness
The paper is technically very sound. The authors provide a formal theoretical foundation for using Value states as update identifiers through Theorem 3.1 and Theorem 3.2, which bound attention output similarity and FFN output divergence. This theoretical justification is a significant advancement over prior empirical heuristics. Theorem 3.4 further ensures that the truncated SVD preserves the topological structure necessary for accurate drift detection. The identification of the "Anisotropy Masking Effect" as the reason for the failure of attention output-based identifiers is a nuanced and well-supported insight.
Presentation
The presentation is clear and well-structured. Figure 4, which decomposes component-wise latency, effectively demonstrates how previous methods were bottlenecked by high-dimensional similarity computations. The workflow diagram in Figure 3 provides a clear conceptual bridge between the three phases of the SPA-Cache layer. The appendices provide rigorous proofs for the theorems presented in the main text.
Significance
The significance of this work is substantial for the growing field of non-causal generative models. DLMs offer flexibility and parallelization potential, but their prohibitive latency has limited production adoption. SPA-Cache bridges this gap, bringing DLM inference speeds closer to AR models without requiring retraining. The compatibility of SPA-Cache with parallel decoding methods (e.g., Fast-dLLM) further amplifies its practical utility.
Originality
The originality lies in the application of SVD for low-dimensional proxy identification in the specific context of DLM caching. While caching and SVD are established techniques, their joint optimization with adaptive layer-wise budget allocation—using a piecewise Gaussian function to model hidden state stability—is a novel contribution to the field.

---

> ### Author Rebuttal · Authors · 2026-03-31
>
> We thank the Reviewer `CQFb` for acknowledging our theoretical and practical contributions and for providing constructive feedback. Responses to specific concerns are provided below. Supplementary tables and figures are available at https://anon0728.github.io/icml-2607-rebuttal/
>
> ---
>
> > Q1. Theorem 3.1 relies on the Stable Attention assumption (bounded total variation of weights). In early stages of diffusion (high noise), does this assumption still hold, and how does it affect the accuracy of the singular proxy?
>
> To validate the Stable Attention assumption, we visualize the total variation $\delta_A = \sum_{j}\|\alpha^{t+1}_{ij}  - \alpha^t\_{ij} \| $ across four representative layers in LLaDA and Dream (Suppl. Fig 6). Our empirical analysis confirms the assumption generally holds, with small $\delta_A < 0.3$ in the majority of cases.
>
> Consistent with intuition, we observed higher variation and occasional spikes in $\delta_A$ during the early decoding steps of LLaDA, particularly in the middle and later layers. To isolate the impact of these initial instabilities on final performance, we implemented a warm-up mechanism where the cache is disabled for the first $N$ decoding steps.
> As shown in Table below (detailed in Suppl. Tab 5), increasing warm-up steps from 0 to 4 yields marginal accuracy gains (<1%) at the cost of reduced throughput.
>
> |  | Warm-up Steps | TPS | Acc |
> | --- | --- | --- | --- |
> | Vanilla | - | 29.67 | 78.62 |
> | Ours * | 0 | 190.73 | 78.24 |
> | Ours | 2 | 185.41 | 78.86 |
> | Ours | 4 | 177.41 | 78.69 |
>
> We conclude that $\delta_A$ spikes in early stages have a negligible impact on overall performance in most scenarios. For high-precision requirements, the warm-up period offers a tunable efficiency-quality trade-off.
>
> ---
>
> > Q2. The adaptive allocation uses a piecewise Gaussian function. How sensitive is the model's performance to the choice of the peak layer and the peak update ratio across different architectures?
>
> The distribution of low-similarity tokens is architecture-dependent, as illustrated in Main Manuscript Fig. 2 and Fig. 6 (Appendix). Consequently, parameters such as the peak layer $l_p$ and peak update ratio $\rho_p$ require architecture-specific configuration. This selection process is straightforward and amenable to automation; in practice, profiling the shift distribution from only 16 samples is generally sufficient and requires approximately two minutes.
>
> To assess the robustness of the Gaussian parameterization, we tested four configurations intentionally shifted from the optimal parameters (Suppl. Fig. 3). Results in Suppl. Tab. 4 demonstrate only marginal accuracy degradation (0.5-2%), confirming that the method is robust to moderate perturbations.
> But random or highly suboptimal settings may lead to significant drops, validating that the piecewise Gaussian function effectively captures the requisite allocation logic and underscores the utility of few-sample profiling.
>
> ---
>
> > Q3. Table 1 shows "Attn. Input" is faster but "Value" is more accurate. Can the authors discuss the trade-off between these two identifier choices in more detail?
>
> The trade-off is driven by computational overhead versus representation purity.
>
> Using the Value state as an identifier requires projecting all tokens into the Value space before computing similarity, which adds latency. In contrast, using the Attn. Input allows for an earlier decision on which tokens to compute, bypassing unnecessary projections and increasing speed.
>
> However, as shown in Main Manuscript Fig. 1 and Suppl. Fig. 1, the Attn. Input is less effective at distinguishing between similar and dissimilar tokens across steps. This is because the input is an entangled representation of Query, Key, and Value states. While $Q$ and $K$ serve as routing features, the final output remains stable if the attended $V$ vectors are semantically equivalent, even if $Q$ or $K$ shifts. Projecting to the Value state filters out these routing fluctuations and preserves only semantic features, resulting in more accurate identification of redundant computations.

---

### Official Review · Reviewer_h4oh · 2026-03-14

**Soundness:** 3
**Presentation:** 3
**Significance:** 3
**Originality:** 2
**Overall Recommendation:** 4
**Confidence:** 4

**Summary:**

This paper tries to address the KV caching inefficiency problem in dLLMs by proposing SPA-Cache, a training-free method that selectively refreshes important token states using a low-rank value-based proxy and a layer-adaptive update budget. Experiments on LLaDA-8B and Dream-7B show better throughput with less accuracy drop.

**Compliance With Llm Reviewing Policy:**

Affirmed.

**Final Justification:**

The rebuttal has clarified distinction with related work like d2cache, with additional clarifications on assumptions behind theoretical underpinning.

**Key Questions For Authors:**

In descending order of priorities and significance:
1. Can you provide a direct comparison with d^2 cache? This is the closest baseline and would materially clarify the paper’s empirical position.
2. How robust is the superiority of singular proxy over other identifiers on other models like Dream, coding benchmarks, beyond the LLaDA-8B on GSM8K ablation?
3. Can you empirically validate the assumptions behind Theorems 3.1 and 3.4, rather than only relying on the theory?
4. What are the quality-efficiency tradeoffs under larger batch sizes, higher temperatures, and distributed inference (e.g. TP/DP)?

**Limitations:**

The paper does discuss some technical limitations, including reduced efficacy at higher temperatures, unchanged TTFT, and added engineering complexity in distributed settings.

But the limitations section would be stronger if it quantified how much accuracy drop it incurs in failure regimes, and more explicitly elaboration on challenges of deploying SPA-Cache in distributed settings.

**Strengths And Weaknesses:**

Strengths

1. The paper targets a real problem in dLLM inference and proposes a training-free design that improves both update identification and layer-wise budget allocation.
2. The method is reasonably well supported by both theory and ablations: the paper gives a formal motivation for using Value states, shows Value is the strongest identifier among several alternatives, and demonstrates that the low-rank singular proxy plus adaptive allocation each contribute measurable gains.
3. Evaluations are good: SPA-Cache is evaluated on two DLMs and seven benchmarks, consistently improves throughput over dLLM-Cache and Fast-dLLM.


Weaknesses

1. The low-rank proxy helps, but the gain over full Value is moderate. Main gain seems to come jointly from the adaptive layer budget.
2. The paper compares against dLLM-Cache and Fast-dLLM, but not against the very close adaptive-caching method like d^2 Cache [1], which also does training-free fine-grained adaptive KV updates. Given the close date d^2 work was published, I consider it more or less a concurrent work but would still be helpful to see a direct comparison and hopefully the authors can provide some further useful insights (on best ways to do adaptive KV updates).
3. the key identifier/rank ablations are mainly on LLaDA-8B+GSM8K, so it is unclear how consistently the same conclusion holds across other model architectures (or dLLMs trained with different methods) like Dream/fast-dLLMv2, and tasks (like coding and conversational tasks).
4. Theoretical justification: the main bounds rely on fairly strong assumptions, especially the bounded-drift condition and the truncated-subspace assumption, and these assumptions are not directly validated empirically.

[1] Yuchu Jiang, Yue Cai, Xiangzhong Luo, Jiale Fu, Jiarui Wang, Chonghan Liu, Xu Yang. d^2 Cache: Accelerating Diffusion-Based LLMs via Dual Adaptive Caching. ICLR 2026.

---

> ### Author Rebuttal · Authors · 2026-03-31
>
> We thank Reviewer `h4oh` for the positive assessment of our motivation and theoretical/practical contributions. Responses to specific concerns are provided below; supplementary Tables/Figures are available at https://anon0728.github.io/icml-2607-rebuttal
>
> ---
>
> > Q1 & W2: Can you provide a direct comparison with d2Cache?
>
> We have included a comparison below (further detailed in Suppl. Tab. 1.) Our method consistently outperforms d2Cache across benchmarks: on LLaDA, we achieve 190 vs. 27 TPS (GSM8k) and 46 vs. 25 TPS (MBPP); on Dream, we achieve 46 vs. 29 TPS (GSM8k) and 115 vs. 42 TPS (MBPP).
>
> LLaDA: GSM8K
>
> |  | TPS | Acc |
> | --- | --- |--- |
> | Vanilla | 29.67 | 78.62 |
> | d2Cache | 26.85 | 76.42 |
> | Ours | 190.73 | 78.24 |
>
> LLaDA: MBPP
>
> |  | TPS | Acc |
> | --- | --- |--- |
> | Vanilla | 5.75 | 39.20 |
> | d2Cache | 25.08 | 39.60 |
> | Ours | 46.12 | 39.00 |
>
> Dream: GSM8K
>
> |  | TPS | Acc |
> | --- | --- | --- |
> | Vanilla | 17.86 | 75.21 |
> | d2Cache | 28.92 | 78.70 |
> | Ours | 45.97 | 77.56 |
>
> Dream: MBPP
>
> |  | TPS | Acc |
> | --- | --- | --- |
> | Vanilla | 36.51 | 57.40 |
> | d2Cache | 41.87 | 57.60 |
> | Ours | 114.85 | 57.60 |
>
> The performance gap stems from architectural compatibility: d2Cache often relies on explicit attention weight computation, which is incompatible with optimized kernels like FlashAttention. Conversely, our method detects shifts using a singular proxy, maintaining compatibility with FlashAttention and achieves higher acceleration at the same or higher accuracy, resulting in a superior efficiency-quality tradeoff.
>
> ---
>
> > Q2 & W3: How robust is the superiority of singular proxy over other identifiers on other models and coding benchmarks?
>
> The Suppl. Tab. 2 & 3 provide ablations on proxy identifiers and singular rank using LLaDA (MBPP, code) and Dream (GSM8K, math).
>
> The results are consistent with our findings in the Main Manuscript Tab. 1 & 5:
>
> - V outperforms Q, K, input, and output identifiers in the efficiency-quality trade-off.
> - Our singular identifier matches the accuracy of the V proxy while further enhancing throughput.
>
> These empirical results verify the robustness of our approach across different models and tasks.
>
> ---
>
> > Q3 & W4: Can you empirically validate the assumptions behind Theorem. 3.1 and 3.4?
>
> We empirically validate these assumptions and conclusions as follows:
>
> - Theorem 3.1 Assumptions
>     1. Bounded Norms Assumption: Suppl. Fig. 5 shows that norms of $h$ and $v$ remain strictly bounded (e.g., LLaDA Layer 4 ranges from 10–20).
>     2. Stable Attention Assumption: Suppl. Fig. 6 shows the variation factor $\delta_A<0.3$ in most cases.
>     3. Bounded Drift Assumption: Suppl. Fig. 4 shows the drift factor $\lambda$ is typically $<3$.
>
> - Theorem 3.1 Conclusion: $1 - S_{\cos}(h^t, h^{t+1}) \leq C (1 - S_{\cos}(v^t, v^{t+1})) + \epsilon$
>     - In Suppl. Fig. 7, we show that setting $C=1.0$ and $\epsilon=0.1$ (in Main Manuscript Eq. 1) covers >99% of the dissimilarity in $h$, confirming the consistency with theoretical conclusion.
>
> - Theorem 3.4, Truncated Subspace Assumption: To validate $h \in \mathrm{span}(V_r)$, we measured the Projection Energy Ratio (PER) in Suppl. Fig. 8. We calculate PER as $\frac{1}{N}\sum_{i=1}^{N}\frac{\|V_r V_r^T h_i\|^2}{\|h_i\|^2}$. A PER near 1 indicates the spanned subspace by $V_r$ effectively captures the energy in $h$ states. For both models, PER exceeds 0.9 across all layers ($r \geq 128$ for LLaDA; $r \geq 32$ for Dream), validating our subspace truncation assumption with suitable ranks and aligning our conclusions in Main Manuscript Tab. 5 and Suppl. Tab. 2 & 3.
>
> - Theorem 3.4 Conclusion: The divergence $S_{\cos}(v_1, v_2) - S_{\cos}(\hat{v}_1, \hat{v}_2)$ is small (and bounded).
>     - As shown in Suppl. Fig. 9, the divergence between the Value identifier and Singular identifier remains concentrated within the $[-0.01, 0.01]$ range, with a maximum divergence below $0.04$. These minimal deviations empirically validate our theoretical results, demonstrating the high fidelity of the singular proxy.
>
> ---
>
> > Q4: What are the quality-efficiency tradeoffs under larger batch sizes, higher temperatures, and distributed inference?
>
> - Batch Size & Distributed Inference: These impact the efficiency-memory trade-off rather than quality. As shown in Suppl. Fig. 10, increasing batch size improves throughput via higher hardware utilization but increases memory overhead. In distributed settings, we primarily employ DP for these 7-8B dLLMs, which follows the same scaling trend as batch size within each GPU.
> - Temperature: Modern LLM deployment typically uses low/zero temperature or greedy decoding. Varying temperature affects output stochasticity in accuracy but does not impact efficiency/throughput.

---

> > ### Author Rebuttal · Reviewer_h4oh · 2026-04-03
> >
> > Thanks for the additional experiments and clarifications. My concerns have been mostly resolved and I will raise my rating.

---

### Decision · Program_Chairs · 2026-04-30

**Decision:**

Accept (regular)

**Comment:**

# Summary

This paper proposes SPA-Cache, a training-free caching framework for Diffusion Language Models (DLMs) that addresses two key bottlenecks in existing approaches: the high computational cost of identifying which tokens require hidden-state updates, and the inefficiency of uniform update budget allocation across layers. The method introduces a low-dimensional singular proxy derived from SVD to cheaply identify update-critical tokens, combined with a layer-adaptive budget allocation strategy. Experiments on LLaDA-8B and Dream-7B across seven benchmarks demonstrate up to 8x throughput improvement over vanilla decoding and 2-4x speedup over prior caching baselines.

# Justification

The paper is well-written and all four reviewers, each with high confidence (4/5), provide positive or constructive assessments. The core contributions are well-supported by evidence: Reviewer CQFb highlights the principled use of SVD for projecting Value states into a compressed subspace, substantially reducing the overhead of similarity checks. Reviewer fjLA agreed that the method achieves up to 8x throughput improvement while maintaining high generation quality across seven benchmarks and this is significant. Reviewer h4oh notes that the practical value of the training-free design and the solid experimental improvements over baselines including Fast-dLLM, dKV-Cache, d2Cache, and dLLM-Cache. Reviewer nV2z, while more reserved, acknowledges that the singular proxy and layer-aware caching budget deliver superior speedup over all existing methods evaluated.

IMHO, this paper makes a clear and timely contribution to DLM inference efficiency. The two proposed mechanisms, the singular proxy for token selection and the adaptive per-layer budget are technically sound and complementary. The experimental evaluation is thorough, covering multiple models (LLaDA-8B, Dream-7B), multiple benchmarks, and direct comparisons to four recent caching baselines.

# Minor weaknesses and suggestions for the camera-ready

Several reviewers note acknowledged limitations that should be addressed more thoroughly in the final version. Reviewer h4oh requests quantification of accuracy degradation in failure regimes (e.g., high-temperature sampling) and a more detailed discussion of distributed deployment challenges. Reviewer CQFb notes that the method’s efficacy depends on the temporal smoothness of hidden state transitions, which should be characterized more precisely. The TTFT (time-to-first-token) is unchanged by SPA-Cache, as noted by Reviewer h4oh. The authors should clarify this limitation explicitly in the final version. Overall, the core contributions are solid and the weaknesses are minor and addressable in revision.